# The EMT transcription factor Snai1 maintains myocardial wall integrity by repressing intermediate filament gene expression

Alessandra Gentile[1], Anabela Bensimon-Brito[1,2†], Rashmi Priya[1,2‡], Hans-Martin Maischein[1], Janett Piesker[3], Stefan Guenther[2,4], Felix Gunawan[1,2*], Didier YR Stainier[1,2*]

[1]Max Planck Institute for Heart and Lung Research, Department of Developmental Genetics, Bad Nauheim, Germany; [2]DZHK German Centre for Cardiovascular Research, Partner Site Rhine-Main, Bad Nauheim, Germany; [3]Max Planck Institute for Heart and Lung Research, Microscopy Service Group, Bad Nauheim, Germany; [4]Max Planck Institute for Heart and Lung Research, Bioinformatics and Deep Sequencing Platform, Bad Nauheim, Germany

*For correspondence:
Felix.Gunawan@mpi-bn.mpg.de (FG);
Didier.Stainier@mpi-bn.mpg.de (DYRS)

Present address: †Aix Marseille Univ, INSERM, Marseille Medical Genetics, Marseille, France; ‡ The Francis Crick Institute, London, United Kingdom

**Abstract** The transcription factor Snai1, a well-known regulator of epithelial-to-mesenchymal transition, has been implicated in early cardiac morphogenesis as well as in cardiac valve formation. However, a role for Snai1 in regulating other aspects of cardiac morphogenesis has not been reported. Using genetic, transcriptomic, and chimeric analyses in zebrafish, we find that Snai1b is required in cardiomyocytes for myocardial wall integrity. Loss of *snai1b* increases the frequency of cardiomyocyte extrusion away from the cardiac lumen. Extruding cardiomyocytes exhibit increased actomyosin contractility basally as revealed by enrichment of p-myosin and $\alpha$-catenin epitope $\alpha$-18, as well as disrupted intercellular junctions. Transcriptomic analysis of wild-type and *snai1b* mutant hearts revealed the dysregulation of intermediate filament genes, including *desmin b* (*desmb*) upregulation. Cardiomyocyte-specific *desmb* overexpression caused increased cardiomyocyte extrusion, recapitulating the *snai1b* mutant phenotype. Altogether, these results indicate that Snai1 maintains the integrity of the myocardial epithelium, at least in part by repressing *desmb* expression.

## Introduction

As the contractile units of the heart, cardiomyocytes (CMs) need to maintain a cohesive tissue-level cytoskeleton to beat synchronously and withstand the high mechanical forces (*Sequeira et al., 2014*; *Gautel and Djinović-Carugo, 2016*). Using zebrafish as a model to analyse CM cytoskeletal organization at single-cell resolution, we searched for candidate transcription factors that regulate CM cytoskeletal and tissue integrity. Amongst the transcription factors involved in cardiac development, we focused on Snai1 (*Nieto, 2002*; *Nieto et al., 2016*), whose orthologues regulate cytoskeletal remodelling and epithelial tissue integrity in *Drosophila* embryos (*Martin et al., 2010*; *Weng and Wieschaus, 2016*) and in mammalian cells in culture (*Wee et al., 2020*). During vertebrate heart formation, Snai1 has been implicated in myocardial precursor migration towards the midline (*Qiao et al., 2014*) and in valve formation (*Tao et al., 2011*), but a role in myocardial wall development, during which an epithelial-to-mesenchymal (EMT)-like process occurs (*Staudt et al., 2014*; *Jiménez-Amilburu et al., 2016*; *Priya et al., 2020*), has not been reported.

## Results

### The transcription factor Snai1b maintains myocardial wall integrity

We focused our attention on one of the zebrafish *snai1* paralogues (*Blanco et al., 2007*), *snai1b*, the knockdown of which has been reported to cause embryonic cardiac defects (*Qiao et al., 2014*). To analyse *snai1b* function, we generated a promoter-less *snai1b* allele (*Figure 1—figure supplement 1A*), which displays almost undetectable levels of *snai1b* mRNA and no transcriptional upregulation of its paralogue (*El-Brolosy et al., 2019*; *Figure 1—figure supplement 1B*). Approximately half of the mutant embryos exhibit cardiac looping defects (*Figure 1—figure supplement 1C–D'*), as reported for *snai1b* morphants (*Qiao et al., 2014*). Upon close examination of the *snai1b* mutant hearts, we observed a new and surprising phenotype leading to a disruption in myocardial wall integrity: CMs extrude away from the cardiac lumen (*Figure 1A–D'*). We found that both heterozygous and homozygous *snai1b* mutant embryos, including ones that display cardiac looping defects, exhibit a significant increase in the number of extruding CMs compared with their wild-type siblings (*Figure 1A–E*, *Figure 1—figure supplement 1E*). The frequency of this CM extrusion is higher in the atrioventricular canal (AVC) (*Figure 1—figure supplement 1F*), where the cells are exposed to stronger mechanical forces from the blood flow and from looping morphogenesis (*Auman et al., 2007*; *Dietrich et al., 2014*; *Bornhorst et al., 2019*). CM extrusion can be observed as early as 48 hours post fertilization (hpf), as well as during larval stages including at 78 (*Figure 1—figure supplement 2A–C*) and 100 (*Figure 1—figure supplement 2D–F*) hpf. By imaging beating hearts over a >18 hours period starting at 52 hpf, we observed that a few extruding CMs in *snai1b* mutants appear to detach from the myocardium and remain in the pericardial cavity for several hours (*Figure 1—figure supplement 1I–K*, *Figure 1—video 1*B); in contrast, we did not observe CMs in the pericardial cavity in wild types (*Figure 1—video 1*A). These results uncover a previously uncharacterized role for Snai1b in maintaining myocardial wall integrity.

For all further analyses, we decided to focus on the *snai1b* mutants displaying apparently unaffected cardiac looping. We first investigated whether the extruding CMs in *snai1b* mutants were apoptotic as dying epithelial cells are frequently removed by extrusion (*Rosenblatt et al., 2001*). However, we did not observe a significant difference in the rate of dying cells, as assessed by terminal deoxynucleotidyl transferase dUTP nick end labelling (TUNEL), between *snai1b*$^{+/+}$ (*Figure 1—figure supplement 3A*) and *snai1b*$^{-/-}$ (*Figure 1—figure supplement 3A', A''*) hearts, indicating that CM extrusion in *snai1b* mutants is not due to cell death.

We next asked whether the defects in myocardial integrity have an impact on cardiac morphology and function. We observed only a small reduction (five cells on average) in CM numbers at 50 hpf (*Figure 1—figure supplement 3B–C*). However, we observed a significant decrease in the number of delaminating CMs in *snai1b*$^{-/-}$ larvae at 78 hpf (*Figure 1—figure supplement 4B*), resulting in fewer trabecular CMs at 100 hpf (*Figure 1—figure supplement 4A, C, D*) compared with wild-type siblings. Furthermore, *snai1b*$^{-/-}$ embryos exhibited an increased CM aspect ratio, as well as reduced apical cell surface and ventricular volume compared with wild-type embryos at 52 (*Figure 1—figure supplement 5A–E*) and 74 (*Figure 1—figure supplement 5F–J*) hpf, indicating a requirement for Snai1b in maintaining CM morphology at both cellular and tissue levels. Although *snai1b*$^{-/-}$ embryos did not exhibit differences in heart rate, ejection fraction, or fractional shortening compared with wild types at 52 hpf (*Figure 1—figure supplement 5K–M*), we observed a significant reduction in all these parameters at 74 hpf (*Figure 1—figure supplement 5N–P*). Taken together, these data suggest that the loss of *snai1b* disrupts cardiac wall morphology, and subsequently cardiac function.

A role for contractility-induced mechanical forces on myocardial wall integrity has recently been reported (*Fukuda et al., 2017*; *Rasouli et al., 2018*; *Fukuda et al., 2019*). Hence, we sought to test whether the loss of cardiac contractility would eliminate the CM extrusion phenotype in *snai1b* mutants, as previously shown for *klf2* mutants (*Rasouli et al., 2018*). We observed that after injecting a *tnnt2a* morpholino (*Sehnert et al., 2002*) to prevent cardiac contraction, the number of extruding CMs in *snai1b* mutants at 50 hpf was significantly reduced (*Figure 1G', H*), and in fact became comparable with that in uninjected *snai1b*$^{+/+}$ embryos (*Figure 1F, H*). These data indicate that mechanical forces due to cardiac contraction are required for the increased frequency of CM extrusion observed in *snai1b* mutants.

To test whether Snai1b plays a cell-autonomous role in promoting myocardial wall integrity, we generated mosaic hearts by cell transplantation (*Figure 1I–L*). We observed that donor-derived

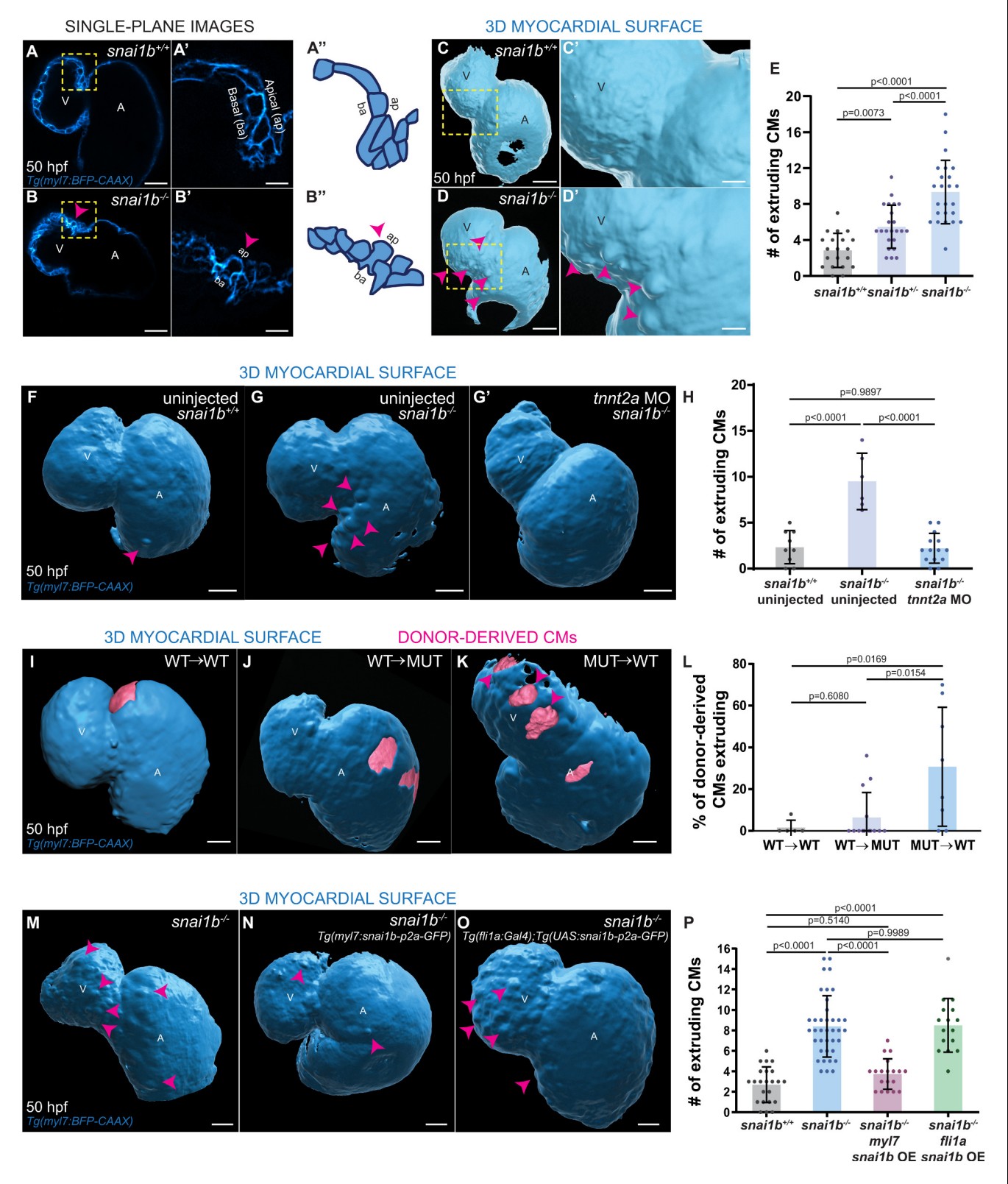

**Figure 1.** Loss of *snai1b* leads to cardiomyocyte (CM) extrusion, disrupting myocardial wall integrity. (**A–B''**) Single-plane images of *Tg(myl7:BFP-CAAX)* *snai1b*<sup>+/+</sup> (**A**) and *snai1b*<sup>-/-</sup> (**B**) hearts at 50 hpf. Close-up of boxed areas (**A', B'**) and schematic (**A'', B''**). (**C–D'**) 3D surface rendering of the myocardium of *Tg(myl7:BFP-CAAX)* *snai1b*<sup>+/+</sup> (**C, C'**) and *snai1b*<sup>-/-</sup> (**D, D'**) embryos at 50 hpf. CM extrusions are clearly observed in *snai1b*<sup>-/-</sup> embryos (magenta arrowheads in **B, B', B'', D, D'**). (**E**) More CMs are extruding in *Tg(myl7:BFP-CAAX)* *snai1b*<sup>-/-</sup> embryos compared with *snai1b*<sup>+/+</sup> and *snai1b*<sup>+/-</sup> siblings at

*Figure 1 continued on next page*

*Figure 1 continued*

50 hpf (*snai1b*$^{+/+}$, n = 20; *snai1b*$^{+/-}$, n = 23; *snai1b*$^{-/-}$, n = 24). (F–H) Blocking cardiac contractions with *tnnt2a* MO leads to a reduced number of extruding CMs in *snai1b*$^{-/-}$ embryos, comparable with uninjected *snai1b*$^{+/+}$ embryos. (F–G') 3D surface rendering of the myocardium of *snai1b*$^{+/+}$ (F) and *snai1b*$^{-/-}$ (G) uninjected embryos and *snai1b*$^{-/-}$ embryos injected with *tnnt2a* MO (G'). (H) Fewer CMs are extruding (magenta arrowheads in G) in *snai1b*$^{-/-}$ embryos injected with *tnnt2a* MO (n = 14) compared with uninjected *snai1b*$^{-/-}$ (n = 6) and *snai1b*$^{+/+}$ (n = 9) embryos at 50 hpf. (I–L) 3D surface rendering of the myocardium showing *snai1b*$^{+/+}$ donor-derived CMs in a *snai1b*$^{+/+}$ (I) or *snai1b*$^{-/-}$ (J) heart, and *snai1b*$^{-/-}$ donor-derived CMs in a *snai1b*$^{+/+}$ heart (K). (L) The percentage of donor-derived CMs that extrude is higher when *snai1b*$^{-/-}$ donor-derived CMs are in *snai1b*$^{+/+}$ hearts (n = 8) than when *snai1b*$^{+/+}$ donor-derived CMs are in *snai1b*$^{+/+}$ (n = 5) or *snai1b*$^{-/-}$ (n = 14) hearts. (M–P) Overexpression of *snai1b* specifically in CMs partially rescues the CM extrusion phenotype in *snai1b*$^{-/-}$ embryos. 3D surface rendering of the myocardium of a *snai1b*$^{-/-}$ embryo (M), and *snai1b*$^{-/-}$ embryo overexpressing *snai1b* under a *myl7* (N) or a *fli1a* (O) promoter. (P) Fewer CMs are extruding (magenta arrowheads) in *snai1b*$^{-/-}$ embryos (n = 19) overexpressing *snai1b* in CMs (N, P) compared with *snai1b*$^{-/-}$ embryos (M, P, n = 38), and this number is comparable to that in *snai1*$^{+/+}$ embryos (n = 24). The number of extruding CMs does not change in *snai1b*$^{-/-}$ embryos (n = 16) when *snai1b* is overexpressed in endothelial cells (*fli1a*) (O, P). Plot values represent means ± S.D.; p-values determined by one-way ANOVA followed by multiple comparisons with Dunn test (E, H, L, P). Scale bars: 20 µm. V: ventricle; A: atrium; ap: apical; ba: basal; n: number of embryos.

The online version of this article includes the following video and figure supplement(s) for figure 1:

**Figure supplement 1.** Generation of *snai1b* mutants.
**Figure supplement 2.** Increased cardiomyocyte (CM) extrusion in *snai1b*$^{-/-}$ larvae.
**Figure supplement 3.** Wild-type like cardiomyocyte (CM) numbers in *snai1b* mutants.
**Figure supplement 4.** *snai1b* mutants exhibit reduced cardiac trabeculation.
**Figure supplement 5.** Altered cardiomyocyte (CM) morphology and function in *snai1b* mutants.
**Figure 1—video 1.** Extruding cardiomyocytes (CMs) in *snai1b*$^{-/-}$ hearts detach from the myocardium and are visible in the pericardial cavity.
https://elifesciences.org/articles/66143#fig1video1

*snai1b*$^{+/+}$ CMs remained integrated in the *snai1b*$^{-/-}$ myocardial wall (***Figure 1J***), whereas donor-derived *snai1b*$^{-/-}$ CMs in a *snai1b*$^{+/+}$ heart were significantly more prone to extrude than their wild-type neighbours (***Figure 1K, L***). Together, these data indicate that *snai1b* is required in a CM-autonomous manner to maintain myocardial wall integrity. Furthermore, we found that CM-specific, but not endothelial-specific, *snai1b* overexpression rescued the *snai1b*$^{-/-}$ CM extrusion phenotype (***Figure 1M–P, Figure 1—figure supplement 1G, H***), further indicating that Snai1b is required in CMs to suppress their extrusion away from the lumen.

## Snai1b limits cardiomyocyte extrusion by regulating the actomyosin machinery

During the process of cardiac trabeculation, some CMs undergo an EMT-like process, lose their apicobasal polarity, and delaminate towards the cardiac lumen (***Staudt et al., 2014***; ***Jiménez-Amilburu et al., 2016***). We wanted to determine whether the extruding CMs in *snai1b* mutants also lose their apicobasal polarity. Notably, we observed that the polarity marker Podocalyxin remained on the apical side of the extruding CMs in *snai1b* mutants (***Figure 2—figure supplement 1A–B''***), suggesting that apicobasal polarity is maintained.

Studies in *Drosophila* embryos and in mammalian cells in culture have revealed the importance of cell extrusion in limiting tissue overcrowding and eliminating dying cells to maintain tissue homeostasis and/or determine cell fate (***Kocgozlu et al., 2016***; ***Wee et al., 2020***). Other experiments have shown that a contractile actomyosin ring around the cell cortex is necessary for their extrusion (***Rosenblatt et al., 2001***; ***Eisenhoffer et al., 2012***; ***Kocgozlu et al., 2016***). Using a monoclonal antibody against the α-catenin epitope α-18 (***Yonemura et al., 2010***), which recognizes the activated conformation of α-catenin, a mechanosensitive protein, and polyclonal antibodies against phosphorylated/activated myosin light chain (p-myosin), we assessed cellular contractility in extruding CMs in *snai1b*$^{+/+}$ and *snai1b*$^{-/-}$ embryos (***Figure 2A–B'', E–F''***). Increased α-catenin epitope α-18 and p-myosin immunofluorescence intensity was observed in the basal side of extruding CMs in *snai1b*$^{-/-}$ (***Figure 2B–C', F–G'***) and *snai1b*$^{+/+}$ (***Figure 2C–C', G–G'***) embryos. As cellular extrusions also involve the rearrangement of cell-cell junctions (***Grieve and Rabouille, 2014***; ***Lubkov and Bar-Sagi, 2014***; ***Teng et al., 2017***), we assessed the localization of the major CM adhesion molecule, N-cadherin (***Bagatto et al., 2006***; ***Cherian et al., 2016***). We observed an overall reduction in N-cadherin levels in the junctions between CMs in *snai1b* mutants compared with those in wild-type siblings

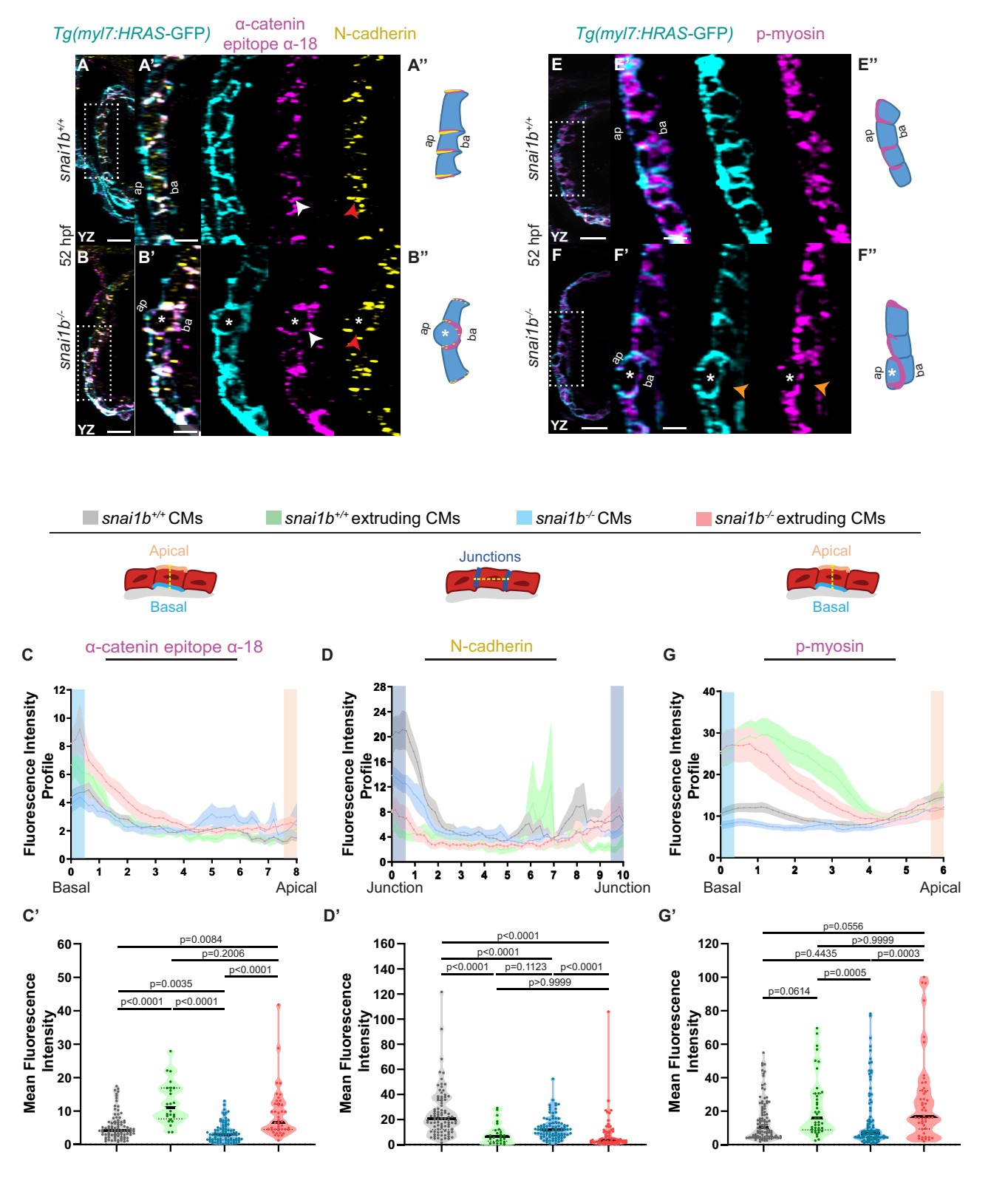

**Figure 2.** Extruding cardiomyocytes (CMs) exhibit changes in actomyosin components. (**A–B″**) Orthogonal projections in the YZ plane of a 52 hpf *snai1b*⁺/⁺ heart (**A**) immunostained for α-catenin epitope α-18, N-cadherin, and GFP compared with a *snai1b*⁻/⁻ sibling heart (**B**). Close-up of boxed areas of *snai1b*⁺/⁺ (**A′**) and *snai1b*⁻/⁻ (**B′**) CMs. Schematics illustrate the localization of activated α-catenin (magenta) in the basal domain of extruding CMs in *snai1b*⁻/⁻ embryos and defects in N-cadherin (yellow) localization in the junctional domain of *snai1b*⁻/⁻ CMs (**A″–B″**). (**C–D′**) Fluorescence intensity

*Figure 2 continued on next page*

Figure 2 continued

profile (FIP) (C–D) and mean fluorescence intensity (mFI) (C'–D') of α-catenin epitope α-18 and N-cadherin immunostaining in 52 hpf *snai1b*$^{+/+}$ and *snai1b*$^{-/-}$ CMs, and in *snai1b*$^{+/+}$ and *snai1b*$^{-/-}$ extruding CMs. The α-catenin epitope α-18 is observed in the basal domain (white arrowhead in B') of extruding CMs (white asterisks in B') in *snai1b*$^{-/-}$ embryos, and a reduction in junctional N-cadherin (red arrowhead in B') is observed in *snai1b*$^{-/-}$ CMs. (FIP α-catenin epitope α-18: *snai1b*$^{+/+}$ CMs, N = 179; *snai1b*$^{+/+}$ extruding CMs, N = 60; *snai1b*$^{-/-}$ CMs, N = 140; *snai1b*$^{-/-}$ extruding CMs, N = 54; mFI α-catenin epitope α-18: *snai1b*$^{+/+}$ CMs, N = 90; *snai1b*$^{+/+}$ extruding CMs, N = 24; *snai1b*$^{-/-}$ CMs, N = 88; *snai1b*$^{-/-}$ extruding CMs, N = 44. FIP N-cadherin: *snai1b*$^{+/+}$ CMs, N = 90; *snai1b*$^{+/+}$ extruding CMs, N = 12; *snai1b*$^{-/-}$ CMs, N = 98; *snai1b*$^{-/-}$ extruding CMs, N = 49; mFI N-cadherin: *snai1b*$^{+/+}$ CMs, N = 90; *snai1b*$^{+/+}$ extruding CMs, N = 25; *snai1b*$^{-/-}$ CMs, N = 92; *snai1b*$^{-/-}$ extruding CMs, N = 70.) (E–F") Representative images of a 52 hpf *snai1b*$^{-/-}$ heart (F) immunostained for p-myosin and GFP compared with a *snai1b*$^{+/+}$ sibling heart (E). Schematics illustrate the basal enrichment of p-myosin (magenta) in extruding CMs in *snai1b*$^{-/-}$ embryos (E"–F"). (G–G') FIP (G) and mFI (G') of p-myosin immunostaining in *snai1b*$^{+/+}$ and *snai1b*$^{-/-}$ CMs, and in *snai1b*$^{+/+}$ and *snai1b*$^{-/-}$ extruding CMs. p-myosin is enriched basally (orange arrowheads in F') in *snai1b*$^{-/-}$ extruding CMs in (white asterisks in F'). (FIP p-myosin: *snai1b*$^{+/+}$ CMs, N = 204; *snai1b*$^{+/+}$ extruding CMs, N = 60; *snai1b*$^{-/-}$ CMs, N = 140; *snai1b*$^{-/-}$ extruding CMs, N = 49; mFI p-myosin: *snai1b*$^{+/+}$ CMs, N = 100; *snai1b*$^{+/+}$ extruding CMs, N = 29; *snai1b*$^{-/-}$ CMs, N = 153; *snai1b*$^{-/-}$ extruding CMs, N = 48). Plot values represent means ± S.E.M. (C, D, G). In the violin plots (C', D', G'), solid black lines indicate median. p-values determined by Kruskal–Wallis test (C', D', G'). Scale bars: 20 µm (A, B, E, F); 2 µm (A', B', E', F'). ap: apical; ba: basal; N: number of CMs. See also *Figure 2—figure supplement 1*.

The online version of this article includes the following figure supplement(s) for figure 2:

**Figure supplement 1.** Apicobasal polarity is maintained in extruding cardiomyocytes (CMs) in *snai1b* mutants.

(*Figure 2A–B", D–D'*), suggesting that Snai1 regulates N-cadherin localization to stabilize actomyosin tension at the junctions, thereby sustaining adhesion between CMs.

## Intermediate filament gene expression is dysregulated in *snai1b*$^{-/-}$ hearts

To further understand how the transcription factor Snai1b is required to maintain myocardial wall integrity, we compared the *snai1b*$^{+/+}$ and *snai1b*$^{-/-}$ cardiac transcriptomes at 48 hpf, a time when CM extrusion is starting to be observed (*Figure 3A*). Since Snai1 primarily acts as a transcriptional repressor (*Baulida et al., 2019*), we focused on the genes upregulated in *snai1b*$^{-/-}$ hearts compared with wild type. In the 339 upregulated genes, gene ontology analysis revealed an enrichment of genes related to the cytoskeleton (*Figure 3—figure supplement 1A*), particularly an upregulation of intermediate filament (IF) genes (*Figure 3B*). Mutations that modify posttranslational modification sites in IF proteins have been associated with cardiomyopathy (*Rainer et al., 2018*), but how IF genes are regulated at the transcriptional level remains poorly understood. Interestingly, the muscle-specific IF gene *desmin b* (*desmb*) was upregulated in *snai1b*$^{-/-}$ hearts (*Figure 3C*), further suggesting that Snai1 modulates CM development cell-autonomously. Desmin is localized to Z-discs and desmosomes within intercalated discs in muscle cells, and an imbalance in Desmin levels is a major cause of cardiomyopathies (*Capetanaki et al., 2015*).

Using quantitative PCR and immunostaining to analyse *desmin* at the mRNA and protein levels, respectively, we first examined the upregulation of *desmb*/Desmin in *snai1b*$^{-/-}$ hearts compared with wild type (*Figure 3D, G–I'*). Notably, extruding *snai1b*$^{-/-}$ CMs exhibit an enrichment of Desmin in their basal domain and a correlative loss of Desmin at intercellular junctions (*Figure 3I–I'*), indicating abnormal Desmin localization. As IFs are known to regulate actomyosin contractility in keratinocytes and astrocytes (*van Bodegraven and Etienne-Manneville, 2020*), these data suggest that basal enrichment of Desmin promotes CM extrusion in *snai1b*$^{-/-}$ hearts.

To further test whether Snai1 represses *desmb* expression, we analysed *desmb* transcript levels upon *snai1b* overexpression. qPCR analysis 4.5 hours after mRNA injection confirmed downregulation of *desmb* transcript levels when *snai1b* was overexpressed (*Figure 3—figure supplement 1B, C*), compared with *gfp* mRNA injected controls. Similarly, qPCR analysis of hearts overexpressing *snai1b* specifically in their CMs showed a reduction of *desmb* transcript levels by 40% (*Figure 3E, F*). ChIP-seq experiments using mouse skeletal myoblasts have shown that murine Snai1 binds to the proximal promoter of *Desmin* (*Soleimani et al., 2012*). Additionally, in silico analysis of zebrafish *desmin* has uncovered potential Snai1b binding sites in the promoter of *desmb*, but not *desma* (*Kayman Kürekçi et al., 2021*). To test whether zebrafish Snai1b can repress the promoter activity of *desmb*, we performed luciferase assays in HEK293T cells. We cloned 800 bp of the proximal promoter of *desmb* upstream of the Firefly *luciferase* gene, and the open reading frame of *snai1b* under a constitutively active promoter. The *desmb* promoter region alone induced transcriptional

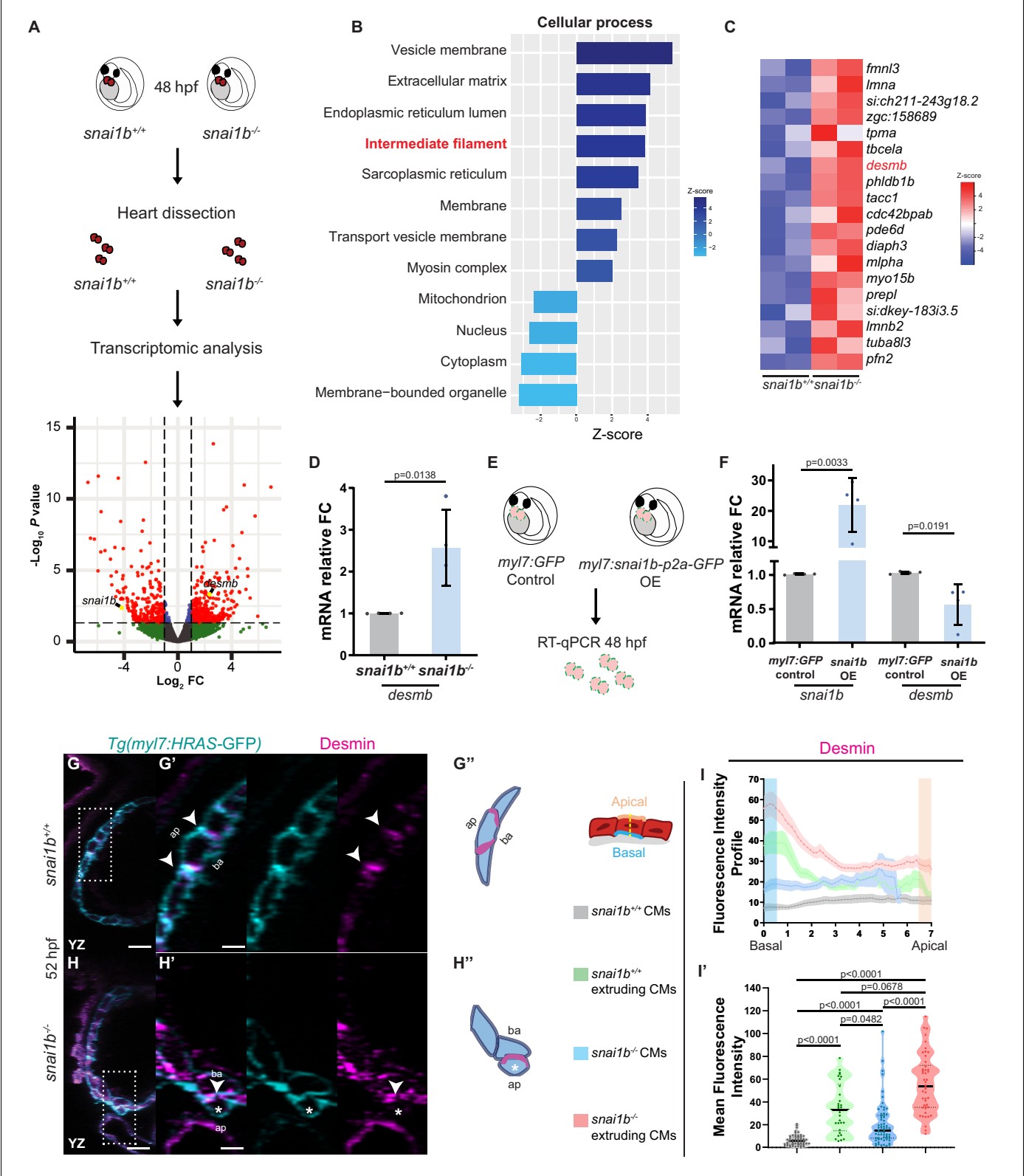

**Figure 3.** Transcriptomic analysis reveals upregulation of intermediate filament genes in *snai1b*⁻/⁻ hearts. (**A**) RNA extracted from 48 hpf *snai1b*⁺/⁺ and *snai1b*⁻/⁻ hearts was used for transcriptomic analysis. (**B**) GO analysis of cellular processes shows enrichment of intermediate filament components in *snai1b*⁻/⁻ hearts. (**C**) Heatmap of a list of upregulated cytoskeletal genes, including *desmb*. (**D**) Relative mRNA levels of *desmb* are significantly increased in *snai1b*⁻/⁻ hearts at 48 hpf; n = 4 biological replicates, 30 hearts each. (**E**) Schematic of *snai1b* overexpression under a *myl7* promoter; *snai1b* and

*Figure 3 continued on next page*

Figure 3 continued

*desmb* mRNA levels analysed at 48 hpf. (F) Relative mRNA levels of *desmb* are significantly reduced in *snai1b* cardiomyocyte (CM)-specific overexpressing hearts at 48 hpf; n = 4 biological replicates, 30 hearts each. (G–H") Orthogonal projections in the YZ plane of a 52 hpf *snai1b*$^{-/-}$ heart (H) immunostained for Desmin and membrane GFP compared with a *snai1b*$^{+/+}$ heart (G). Close-up of boxed areas of *snai1b*$^{+/+}$ (G') and *snai1b*$^{-/-}$ (H') CMs. Schematics (Desmin in magenta) illustrate the basal enrichment of Desmin in extruding CMs in *snai1b*$^{-/-}$ embryos (G"–H"). (I–I') Fluorescence intensity profile (FIP) (I) and mean fluorescence intensity (mFI) (I') of Desmin in *snai1b*$^{+/+}$ and *snai1b*$^{-/-}$ CMs, and in *snai1b*$^{+/+}$ and *snai1b*$^{-/-}$ extruding CMs. Desmin immunostaining is observed throughout the *snai1b*$^{-/-}$ myocardium, with an enrichment in the basal domain (white arrowheads in H'–G') in extruding CMs (white asterisks in H'). (FIP: *snai1b*$^{+/+}$ CMs, N = 49; *snai1b*$^{+/+}$ extruding CMs, N = 41; *snai1b*$^{-/-}$ CMs, N = 45; *snai1b*$^{-/-}$ extruding CMs, N = 41; mFI: *snai1b*$^{+/+}$ CMs, N = 56; *snai1b*$^{+/+}$ extruding CMs, N = 30; *snai1b*$^{-/-}$ CMs, N = 65; *snai1b*$^{-/-}$ extruding CMs, N = 46). Plot values represent means ± S.D. (D, F) or mean ± S.E.M. (I). In the violin plot (I'), solid black lines indicate median. p-Values determined by Student's t-test (D, F) or Kruskal–Wallis test (I'). Scale bars: 20 µm (G, H); 2 µm (G', H'). ap: apical; ba: basal; n: number of embryos; N: number of CMs; FC: fold change. All Ct values are listed in *Supplementary file 2*. See also *Figure 3—figure supplement 1*.

The online version of this article includes the following figure supplement(s) for figure 3:

**Figure supplement 1.** Snai1b regulates *desmb* expression.

activation of Luciferase compared with control. However, co-expression of Snai1b led to a significant reduction of the Luciferase signal (*Figure 3—figure supplement 1D*), suggesting that Snai1b can repress the promoter activity of *desmb*. Taken together, these data suggest that Snai1b regulates *desmb* transcription.

## *desmb* overexpression in cardiomyocytes promotes their extrusion

Both loss (*Taylor et al., 2007*; *Ramspacher et al., 2015*) and gain (*Chen et al., 2018*) of Desmin expression have been associated with cardiac defects. Thus, we asked whether an imbalance in *desmb* expression could lead to CM extrusion by overexpressing *desmb* mosaically in CMs. We observed that *desmb* overexpressing CMs were more prone to extrude compared with *gfp* overexpressing CMs (*Figure 4A–C*), suggesting that IFs are needed at their endogenous levels to maintain myocardial wall integrity. We hypothesized that increased Desmin levels induce CM extrusion by disrupting desmosome organization leading to compromised cell-cell adhesion and/or by increasing cell contractility basally. We first used electron microscopy to analyse desmosomes at the ultrastructural level, but observed no obvious defects in *snai1b*$^{-/-}$ CMs compared with wild type (*Figure 4—figure supplement 1A–D*). This result is consistent with a previous study that shows intact desmosomes in extruding epithelial cells (*Thomas et al., 2020*). To test whether overexpression of Desmin in CMs was associated with increased cell contractility, we performed immunostaining on *desmb* overexpressing hearts using α-catenin epitope α-18, p-myosin, and Desmin antibodies. *desmb* overexpressing CMs exhibited a basal enrichment of Desmin (*Figure 4H–I'*), as well as of the activated actomyosin factors α-catenin epitope α-18 and p-myosin (*Figure 4D–G'*). As we observed in *snai1b*$^{-/-}$ CMs, *desmb* overexpressing CMs also exhibited reduction of N-cadherin at the junctions compared with control (*Figure 4—figure supplement 2A–B'*). Taken together, these data show that increasing *desmb* expression in CMs compromises their adhesion (reduced N-cadherin) and increases their basal actomyosin contractility (increased α-catenin epitope α-18 and p-myosin), recapitulating *snai1b* mutant phenotypes.

## Discussion

A role for Snai1 in cell extrusion has been reported in *Drosophila* embryos as well as in mammalian cells in culture. During *Drosophila* gastrulation, Snai1 promotes the medio-apical pulsations of contractile Myo-II that drive apical constriction (*Martin et al., 2009*; *Martin et al., 2010*; *Mitrossilis et al., 2017*). However, the transcriptional targets of Snai1 that promote cellular contractility in this system remain unknown. Recent in vitro studies have reported Snai1-mediated upregulation of active RhoA, leading to increased cortical actomyosin activity and apical extrusion (*Wee et al., 2020*). Here, our work uncovers a previously unsuspected role for the EMT-inducing factor Snai1 in limiting CM extrusions by regulating IF gene expression. We show that the CM extrusions in *snai1b* mutants are associated with increased accumulation of actomyosin basally, providing more evidence for a role of Snai1 in regulating cell contractility through actin networks. This function appears to be partly independent of Snai1's role in EMT as no obvious changes in Podocalyxin

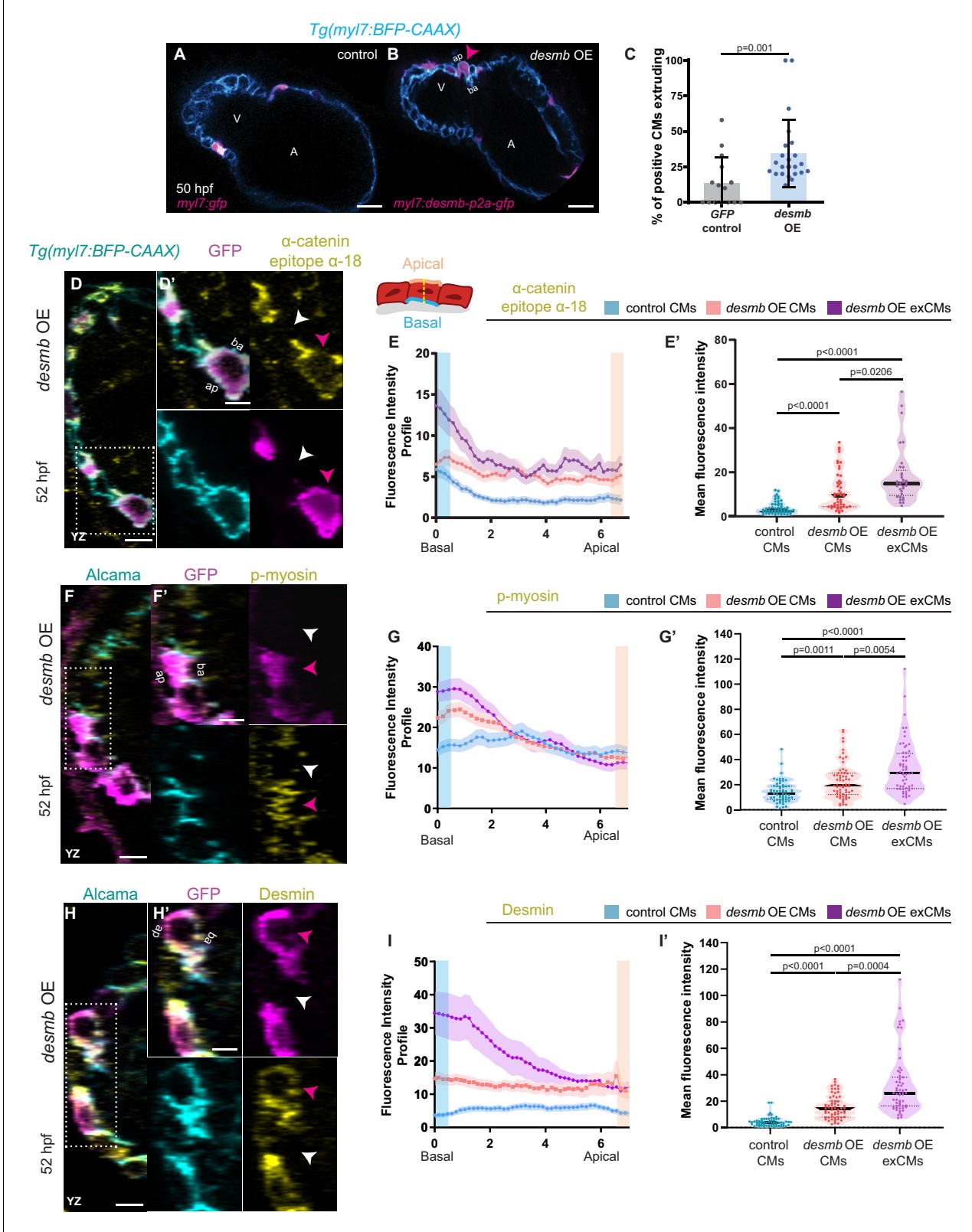

**Figure 4.** *desmb* overexpression in cardiomyocytes (CMs) induces their extrusion. (**A, B**) Single-plane images of *snai1b⁺/⁺* embryos injected with *myl7: GFP* (**A**) or with *myl7:desmb-p2a-GFP* (**B**) at 50 hpf. (**C**) A higher percentage of CMs extrude when overexpressing *desmb* (n = 23) compared with control (n = 15) (magenta arrowheads in **B, B'**). (**D–D', F–F', H–H'**) Orthogonal projections in the YZ plane of hearts from 52 hpf embryos injected with *myl7:desmb-p2a-GFP* and immunostained for α-catenin epitope α-18, GFP, and BFP (**D–D'**), p-myosin, GFP, and Alcama (**F–F'**), or Desmin, GFP, and

*Figure 4 continued on next page*

## eLife Short report

Cell Biology | Developmental Biology

localization were observed. Our data show the requirement of Snai1 in maintaining epithelial tissue integrity in a vertebrate organ and add to the growing evidence that Snai1 has EMT-independent roles in epithelial tissues.

Alcama (**H–H'**). Close-up of boxed areas of *desmb*-overexpressing and adjacent wild-type CMs (**D', F', H'**). (**E–E', G–G', I–I'**) Fluorescence intensity profile (FIP) (**E, G, I**) and mean fluorescence intensity (mFI) (**E', G', I'**) of α-catenin epitope α-18 (**E–E'**), p-myosin (**G–G'**), and Desmin (**I–I'**) in CMs that overexpress *desmb* (magenta arrowheads) and CMs that do not overexpress *desmb* (white arrowheads). α-catenin epitope α-18, p-myosin, and Desmin immunostaining signals are enriched in the basal domain in *desmb* overexpressing CMs. FIP: *desmb* OE CMs, (**E**) N = 132, (**G**) N = 120, (**I**) N = 88; *desmb* OE extruding CMs, (**E**) N = 47, (**G**) N = 54, (**I**) N = 49; control CMs, (**E**) N = 153, (**G**) N = 133, (**I**) N = 86. mFI: *desmb* OE CMs, (**E'**) N = 49, (**G'**) N = 65, (**I'**) N = 63; *desmb* OE extruding CMs, (**E'**) N = 33, (**G'**) N = 60, (**I'**) N = 59; control CMs, (**E'**) N = 55, (**G'**) N = 62, (**I'**) N = 64. Plot values represent means ± S.D. (**C**) or means ± S.E.M. (**E, G, I**). In the violin plots (**E', G', I'**), solid black lines indicate median. p-values determined by Mann–Whitney *U* (**C**) or Kruskal–Wallis (**E', G', I'**) test. Scale bars: 20 µm (**A–B', D, F, H**); 2 µm (**D', F', H'**). V: ventricle; A: atrium; ap: apical; ba: basal; n: number of embryos; N: number of CMs. See also *Figure 4—figure supplements 1–3*.

**Figure supplement 1.** Desmosome organization is maintained in *snai1b* mutants.
**Figure supplement 2.** *desmb* overexpression in cardiomyocytes (CMs) leads to reduced junctional N-cadherin.
**Figure supplement 3.** Model.

Furthermore, we report a previously uncharacterized function of Snai1 in regulating *desmin* expression and find that an increase in Desmin levels perturbs tissue integrity. Although IFs including vimentin (*Kajita et al., 2014*) and keratin (*Kadeer et al., 2017*; *Thomas et al., 2020*) have been reported to accumulate at the interface between extruding cells and their neighbours, our study provides evidence that increased Desmin levels are correlated with mislocalization of the actomyosin machinery in the basal domain of extruding cells. These data are consistent with previous findings that IFs can regulate the actomyosin network, with factors such as vimentin binding to actin and modulating RhoA activity (*Jiu et al., 2017*), and keratin binding to myosin (*Kwan et al., 2015*). In addition to a role for Desmin in maintaining nuclear membrane architecture in CMs (*Heffler et al., 2020*), our results shed light on the function of Desmin in maintaining myocardial wall integrity.

Our results also uncover the requirement of Snai1 and the correct levels of Desmin in maintaining myocardial wall integrity under contraction-induced mechanical pressure. Cardiac contraction is essential in patterning the cardiac tissue: without a heartbeat, cardiac valves and the trabecular network fail to form (*Granados-Riveron and Brook, 2012*; *Collins and Stainier, 2016*). Our results further indicate that without a strong intracellular cytoskeletal network regulated by Snai1, the heartbeat-induced mechanical forces can lead to an increase in CM extrusion and loss of myocardial wall integrity. While it has been shown that an increase in cell contractility due to changes in morphology, adhesion, or cell density drives cell extrusion (*Eisenhoffer et al., 2012*; *Levayer et al., 2016*; *Kocgozlu et al., 2016*; *Saw et al., 2017*; *Miroshnikova et al., 2018*; *Campinho et al., 2020*; *Priya et al., 2020*), we present evidence that external mechanical forces contribute to non-apoptotic CM extrusion during cardiac development, and that actomyosin and IF cytoskeletal regulation prevent CM extrusion.

In conclusion, our findings uncover molecular mechanisms that suppress cell extrusion in a tissue under constant mechanical pressure and show a multifaceted, context-dependent role for Snai1 in promoting EMT (*Nieto et al., 2016*), and also in maintaining tissue integrity during vertebrate organ development (*Figure 4—figure supplement 3*).

## Materials and methods

### Key resources table

| Reagent type (species) or resource | Designation | Source or reference | Identifiers | Additional information |
|---|---|---|---|---|
| Antibody | Anti-tRFP (rabbit polyclonal) | Evrogen | RRID:AB_2571743 | IF(1:200) |

*Continued on next page*

*Continued*

| Reagent type (species) or resource | Designation | Source or reference | Identifiers | Additional information |
|---|---|---|---|---|
| Antibody | Anti-GFP (chicken polyclonal) | AvesLab | RRID:AB_10000240 | IF(1:800) |
| Antibody | Anti-N-cadherin (rabbit polyclonal) | Abcam | RRID:AB_444317 | IF(1:250) |
| Antibody | Anti-p-myosin (rabbit polyclonal) | Abcam | RRID:AB_303094 | IF(1:200) |
| Antibody | Anti-Desmin (rabbit polyclonal) | Sigma | RRID:AB_476910 | IF(1:100) |
| Antibody | Anti-α-catenin epitope α-18 (rat monoclonal) | Gift from Prof. Akira Nagafuchi | | IF(1:300) |
| Antibody | Anti-Alcama (mouse monoclonal) | DSHB | RRID:AB_531904 | IF(1:50) |
| Antibody | Alexa Fluor 488 Goat anti Chicken IgG (H + L) | Thermo Fisher Scientific | RRID:AB_142924 | IF(1:500) |
| Antibody | Alexa Fluor 647 Goat anti Rabbit IgG (H + L) | Thermo Fisher Scientific | RRID:AB_141663 | IF(1:500) |
| Antibody | Alexa Fluor 647 Goat anti Rat IgG (H + L) | Thermo Fisher Scientific | RRID:AB_141778 | IF(1:500) |
| Antibody | Alexa Fluor 568 Goat anti Rabbit IgG (H + L) | Thermo Fisher Scientific | RRID:AB_2534123 | IF(1:500) |
| Antibody | Alexa Fluor 568 Goat anti Rat IgG (H + L) | Thermo Fisher Scientific | RRID:AB_2534121 | IF(1:500) |
| Chemical compound, drug | Agarose, low gelling temperature | Sigma | Cat# A9414-25g | |
| Chemical compound, drug | Bovine serum albumin | Sigma | Cat# A-9418 | |
| Chemical compound, drug | Chloroform | Merck | Cat# 102445 | |
| Other | DAPI | Sigma | Cat# D954 | (1 µg/mL) |
| Chemical compound, drug | Dimethyl sulfoxide (DMSO) | Sigma | Cat# D8418 | |
| Chemical compound, drug | DMEM(1X)+Glutamax | Thermo Fisher Scientific | Cat# 31966-021 | |
| Chemical compound, drug | DyNAmo ColorFlash SYBR Green qPCR Mix | Thermo Fisher Scientific | Cat# F416S | |
| Chemical compound, drug | Ethanol, undenatured, absolute | Serva | Cat# 11093.01 | |
| Chemical compound, drug | FBS superior | Biochrom | Cat# S0615 | |
| Chemical compound, drug | Glycine | Sigma | Cat# 50046 | |
| Chemical compound, drug | 2-Propanol | Roth | Cat# 6752.4 | |
| Chemical compound, drug | Lipofectamine 3000 Transfection Reagent | Thermo Fisher Scientific | L3000001 | |
| Chemical compound, drug | Methanol | Roth | Cat# 4627.5 | |
| Chemical compound, drug | Normal Goat Serum | Thermo Fisher Scientific | Cat# 16210072 | |
| Chemical compound, drug | Paraformaldehyde | Sigma | Cat# P6148 | |

*Continued*

| Reagent type (species) or resource | Designation | Source or reference | Identifiers | Additional information |
|---|---|---|---|---|
| Chemical compound, drug | Phosphate buffered saline (PBS) | Sigma | Cat# P4417 | |
| Recombinant DNA reagent | pT3TS-nCas9n (plasmid) | Addgene | Cat# 46757 | |
| Recombinant DNA reagent | pCS2z vector (plasmid) | Addgene | Cat# 62214 | |
| Recombinant DNA reagent | pCMV-Tol2 (plasmid) | Addgene | Cat# 31823 | |
| Recombinant DNA reagent | pGl4.14-luc; SV40: hRLuc (plasmid) | *Bensimon-Brito et al., 2020* | | |
| Chemical compound, drug | Sodium citrate monobasic | Sigma | Cat# 71497-1KG | |
| Chemical compound, drug | Triton X-100 | Sigma | Cat# X-100 | |
| Chemical compound, drug | TRIzol Reagent | Thermo Fisher Scientific | Cat# 15596026 | |
| Chemical compound, drug | Tween 20 | Sigma | Cat# P1379 | |
| Commercial assay or kit | Dual-Luciferase Reporter Assay System | Promega | Cat# E1910 | |
| Commercial assay or kit | In Situ Cell Death Detection Kit, Fluorescein | Roche | 11684795910 | |
| Commercial assay or kit | Maxima First Strand cDNA kit | Thermo Fisher Scientific | Cat# K1641 | |
| Commercial assay or kit | MegaShortScript T7 Transcription Kit | Thermo Fisher Scientific | Cat# AM1354 | |
| Commercial assay or kit | MegaScript T3 Transcription Kit | Thermo Fisher Scientific | Cat# AM1348 | |
| Commercial assay or kit | mMESSAGE mMACHINE T7 Transcription Kit | Thermo Fisher Scientific | Cat# AM1344 | |
| Commercial assay or kit | mMESSAGE mMACHINE T3 Transcription Kit | Thermo Fisher Scientific | Cat# AM1348 | |
| Commercial assay or kit | RNA Clean and Concentrator Kit | Zymo Research | Cat# R1013 | |
| Cell line (*Homo sapiens*) | HEK-293T | ATCC | Cat# CRL-3216 | RRID:CVCL_0063 |
| Strain, strain background (*Danio rerio*) | *Tg(myl7:BFP- CAAX)*[bns193] | *Guerra et al., 2018* | ZFIN:bns193 | |
| Strain, strain background (*Danio rerio*) | *Tg(myl7:H2B-EGFP)*[zf521Tg] | *Mickoleit et al., 2014* | ZFIN:zf521Tg | |
| Strain, strain background (*Danio rerio*) | *Tg(myl7:mVenus-gmnn)*[ncv43Tg] | *Jiménez-Amilburu et al., 2016* | ZFIN:ncv43Tg | |
| Strain, strain background (*Danio rerio*) | *Tg(−0.2myl7:snai1b-p2a-GFP)* [bns555] | This paper | ZFIN:bns555 | |
| Strain, strain background (*Danio rerio*) | *Tg(−0.2myl7:EGFP-podocalyxin)* [bns10] | *Jiménez-Amilburu et al., 2016* | ZFIN:bns10 | |

*Continued on next page*

*Continued*

| Reagent type (species) or resource | Designation | Source or reference | Identifiers | Additional information |
|---|---|---|---|---|
| Strain, strain background (*Danio rerio*) | *Tg(fli1a:Gal4)*[ubs4] | *Zygmunt et al., 2011* | ZFIN:ubs4 | |
| Strain, strain background (*Danio rerio*) | *Tg(UAS:snai1b-p2a-GFP)* [bns442] | This paper | ZFIN:bns442 | |
| Strain, strain background (*Danio rerio*) | *Tg(myl7:EGFP-Hsa.HRAS)*[s883Tg] | *D'Amico et al., 2007* | ZFIN:s883Tg | |
| Strain, strain background (*Danio rerio*) | *snai1b*[bn351] mutant | This paper | ZFIN:bns351 | |
| Sequence-based reagent | qPCR | This paper | Table S1 | |
| Sequence-based reagent | Genotyping | This paper | Table S1 | |
| Sequence-based reagent | PCR | This paper | Table S1 | |
| Software, algorithm | FiJi ImageJ 1.53 c | *Schindelin et al., 2012* | RRID:SCR_002285 | |
| Software, algorithm | GraphPad Prism 6 | GraphPad | RRID:SCR_002798 | |
| Software, algorithm | Imaris, version 8.4.0 | Bitplane | RRID:SCR_007370 | |
| Software, algorithm | Zen Digital Imaging | Carl Zeiss Microscopy | RRID:SCR_013672 | |

## Zebrafish husbandry

Zebrafish husbandry was performed in accordance with institutional (MPG) and national (German) ethical and animal welfare regulation. Larvae were raised under standard conditions. Adult zebrafish were maintained in 3.5 L tanks at a stock density of 10 zebrafish/L with the following parameters: water temperature: 27–27.5°C; light:dark cycle: 14:10; pH: 7.0–7.5; conductivity: 750–800 µS/cm. Zebrafish were fed 3–5 times a day, depending on age, with granular and live food (*Artemia salina*). Health monitoring was performed at least once a year. All embryos used in this study were raised at 28°C and staged at 75% epiboly for synchronization.

All procedures performed on animals conform to the guidelines from Directive 2010/63/EU of the European Parliament on the protection of animals used for scientific purposes and were approved by the Animal Protection Committee (Tierschutzkommission) of the Regierungspräsidium Darmstadt (reference: B2/1218).

## Zebrafish lines

The following lines were used in this study: *Tg(myl7:BFP-CAAX)bns193* (*Guerra et al., 2018*); *Tg(myl7:H2B-EGFP)zf521* (*Mickoleit et al., 2014*); *Tg(myl7:mVenus-gmnn)ncv43* (*Jiménez-Amilburu et al., 2016*); *Tg(−0.2myl7:EGFP-podocalyxin)bns10* (*Jiménez-Amilburu et al., 2016*); *Tg(fli1a:Gal4)ubs4* (*Zygmunt et al., 2011*); *Tg(myl7:EGFP-Hsa.HRAS)s883* (*D'Amico et al., 2007*); *Tg(UAS:snai1b-p2a-GFP)bns442* (this study); *Tg(−0.2myl7:snai1b-p2a-GFP)bns555* (this study); and *snai1b*[bns351] (this study).

## Generation of transgenic lines

To generate the *snai1b* overexpression lines, the full coding sequence was amplified by PCR using the following primers: forward – 5′-ATGCCACGCTCATTTCTTGT-3′ and reverse – 5′-GAGCGCCG-GACAGCAGCC-3′. The 765 bp amplicon was cloned into pT2-UAS and into an iSce-I plasmid

downstream of a −0.2myl7 promoter and upstream of a P2A linker and GFP. All cloning experiments were performed using ColdFusion Cloning (System Biosciences). The plasmids were then injected into AB embryos at the one-cell stage (25 pg/embryo) together with *Tol2* mRNA (25 pg/embryo) to generate *Tg(UAS:snai1b-p2a-GFP)* and *Tg(−0.2myl7:snai1b-p2a-GFP)*, respectively.

## Generation of the *snai1b$^{bns351}$* allele

The *snai1b* mutant allele was generated using the CRISPR/Cas9 technology. Guide RNA (gRNA) sequences were designed using the CRISPOR program (http://crispor.tefor.net/). To generate a promoter-less allele, two gRNAs were designed: one targeting the proximal promoter (5′-GTCTATAAG TGGCGCAG-3′) and another targeting exon 1, immediately after the sequence encoding the SNAG domain (5′-GTAGTTTGGCTTCTTGT-3′), resulting in a deletion of 1300 bp. The gRNAs were transcribed using a MegaShortScript T7 Transcription Kit (Thermo Fisher Scientific). *cas9* mRNA was transcribed using an mMESSAGE mMACHINE T3 Transcription Kit (Thermo Fisher Scientific) using pT3TS-nCas9n as a template. The RNAs were purified with an RNA Clean and Concentrator Kit (Zymo Research). gRNAs (~12.5 pg/embryo for each gRNA) and *cas9* mRNA (~300 pg/embryo) were co-injected at the one-cell stage. High-resolution melt analysis (HRMA) was used to determine the efficiency of the gRNAs. For genotyping, a reverse primer (5′-AATTTCACTCTCACCAGTCTGA-3′) was combined with a forward primer in the promoter region (5′-ACCTTCTTGTTGTGAGGCGA-3′) to detect the mutant allele, and with a forward primer in exon 1 (5′-ATGCCACGCTCATTTCTTGTCAA-3′) to detect the wild-type allele.

## Overexpression of *snai1b*

A full-length *snai1b* cDNA was cloned from 48 hpf cDNA into the pCS2+ vector (Addgene). In vitro transcription using a mMESSAGE mMACHINE T7 Transcription Kit (Thermo Fisher Scientific) generated *snai1b* mRNA. Wild-type embryos were injected at the one-cell stage with 25 pg of *snai1b* or *gfp* mRNA. RNA from 40 4.5 hpf embryos was extracted using a standard phenol/chloroform protocol.

## Overexpression of *desmb*

To generate the *desmb* overexpression plasmid, the full coding sequence was amplified by PCR using the following primers: forward – 5′-ATGAGCCACTCTTATGCCAC-3′ and reverse – 5′-CA TGAGGTCCTGCTGGTG-3′. The 1419 bp amplicon was cloned into a iSce-I plasmid downstream of a −0.2myl7 promoter and upstream of a P2A linker and GFP. All cloning experiments were performed using ColdFusion Cloning (System Biosciences). The plasmid was then injected into *Tg(myl7: BFP-CAAX)* embryos at the one-cell stage (25 pg/embryo) together with *Tol2* mRNA (25 pg/embryo) to obtain mosaic expression.

## Immunohistochemistry

Embryos were collected, treated with 1-phenyl-2-thiourea (PTU) at 24 hpf to prevent pigmentation, and fixed in 4% PFA for 2 hours at room temperature, after stopping the heart with 0.4% Tricaine to prevent it from collapsing during fixation. After exchanging the fixative with PBS/0.1% Tween washes, yolks were removed using forceps, incubated in 0.1 M glycine for 10 min, and then washed with PBS/1% BSA/1% DMSO/0.5% Triton-X (PBDT), and blocked with PBDT/10% goat serum before incubating in primary antibody at 4°C overnight. The embryos were washed in PBDT and incubated in secondary antibody for 2 hours at room temperature, then incubated with DAPI (2 μg/mL) for 10 min and washed with PBS/0.1% Tween.

Primary antibodies used were GFP (Abcam, 1:800 dilution); N-cadherin (Abcam, 1:250 dilution); p-myosin (Abcam, 1:200); tRFP (Evrogen, 1:200 dilution); Desmin (Sigma, 1:100); and Alcama (DSHB ZN-8, 1:50). α-catenin epitope α-18 (1:300) antibody was a generous gift from Prof. Akira Nagafuchi. Secondary antibodies (1:500 dilution) used were Alexa Fluor 568, Alexa Fluor 488, and Alexa Fluor 647 (Thermo Fisher Scientific).

## Imaging

Confocal microscopes were used to image stopped hearts. Embryos were mounted in 1% low-melting agarose with 0.2% Tricaine, and the stopped hearts were imaged using a Zeiss LSM700 or

LSM880 confocal microscope with a 20× or 40× dipping lens. Fixed embryos were mounted in 1% low-melting agarose and were imaged using a Zeiss LSM700 or LSM880 confocal microscope with a 20× or 40× dipping lens, and genotyped afterwards.

### Heart rate, ventricular ejection fraction, and ventricular fractional shortening quantification

Live imaging of beating hearts was performed using a Zeiss Spinning Disk confocal microscope. Zebrafish at 48, 78, and 100 hpf were mounted in 2% low-melting agarose without Tricaine. 20–30 s movies were recorded with 5 ms exposure. Light intensity and duration were kept to a minimum to avoid light-induced twitching. Kymographs were generated using ImageJ, and ventricular ejection fraction and ventricular fractional shortening were quantified with ImageJ.

### TUNEL assay

Embryos at 50 hpf were fixed in 4% PFA for 2 hours at room temperature, washed in PBS/0.1% Tween, and manually deyolked with insulin needles. Samples were dehydrated and stored in 100% MeOH at −20°C overnight. After rehydration, embryos were processed for antibody staining (Evrogen, tRFP 1:200) . Subsequently, samples were permeabilized with 0.1% sodium citrate in PBS for 2 min on ice. After washes in PBS/0.3% Triton-X, embryos for the positive control were incubated for 15 min at 37°C with DNAaseI. All the embryos were incubated for 1 hour at 37°C in the TUNEL solution (In Situ Cell Death Detection Kit Fluorescein, Roche). After washes, embryos were mounted for imaging.

### Quantitative PCR analysis

Dissected hearts were homogenized in TRIzol (Thermo Fisher Scientific) using a NextAdvance Bullet Blender Homogenizer, followed by standard phenol/chloroform extraction. At least 500 ng of total RNA was used for reverse transcription using a Maxima First Strand cDNA synthesis kit (Thermo Fisher Scientific). For all experiments, DyNAmo ColorFlash SYBR Green qPCR Mix (Thermo Fisher Scientific) was used on a CFX connect Real-time System (Bio-Rad) with the following program: pre-amplification 95°C for 7 min, amplification 95°C for 10 s and 60°C for 30 s (39 cycles), melting curve 60–92°C with increment of 1°C each 5 s. Each point in the dot plots represents a biological replicate from three technical replicates. Gene expression values were normalized using the housekeeping gene *rpl13a* and fold changes were calculated using the $2^{-\Delta\Delta Ct}$ method; all Ct values are listed in *Supplementary file 2*. Primer sequences can be found in Table S1.

### Blastomere transplantations and morpholino injections

Cells obtained from mid-blastula stage donor embryos were transplanted along the blastoderm margin of age-matched host embryos. A *tnnt2a* ATG-MO was injected into the yolk at the one-cell stage at 0.3 ng per embryo. The embryos were then imaged at 52 hpf.

### Image analysis

All immunostainings were analysed in the YZ orthogonal plane to better visualize CM extrusion. The line scan function of Fiji was used to quantify fluorescence intensity at the junctional and basal domains. To visualize the fluorescence profile of N-cadherin immunostaining, a line of uniform thickness was drawn from junction to junction in adjacent CMs. To analyse the localization of α-catenin epitope α-18, p-myosin, and Desmin, a line of uniform thickness was drawn from the basal to the apical domain of CMs. Asymmetry in fluorescence intensity appears due to variable CM length. To assess fluorescence intensity, the mean grey values were used, drawing a line of uniform thickness at the junctional (N-cadherin) or basal (α-catenin epitope α-18, p-myosin, and Desmin) domain of CMs. Images were processed and analysed with Fiji. A background subtraction of rolling ball radius 20 was applied, followed by a mean filter of radius 1. Brightness and contrast were adjusted to remove any background fluorescence. Apical cell surface and aspect ratio were quantified using the line function of ImageJ.

The total number of CMs was counted using the Spots function, and 3D cardiac surface rendering and ventricular volume quantification were obtained with the Surfaces function of the Imaris Bitplane Software.

## Luciferase assay and plasmids

To generate the plasmid with a zebrafish *desmb* promoter driving Firefly Luciferase expression (pGl4.14-luc; SV40:hRLuc) (*Bensimon-Brito et al., 2020*), we cloned 800 bp of the promoter region of *desmb* using the following primers: forward – 5′-GAAAGCATAGTCTGCTTTCTCG-3′ and reverse – 5′-GAGCGCCGGACAGCAGCC-3′.

The zebrafish *snai1b* coding sequence was inserted downstream of the CMV promoter in the pCMV-Tol2 plasmid (Addgene). The full-length *snai1b* coding sequence was amplified using the following primers: forward – 5′-ATGCCACGCTCATTTCTTGT-3′ and reverse – 5′-GAGCGCCGGACAG-CAGCCGGAC3′. Per well in a 24-well plate, HEK-293T cells were transfected with 200 ng of the luciferase plasmid and 200 ng of pCMV-*snai1b* or the empty plasmid as control, as well as 1.5 µL Lip-ofectamine 3000 Transfection Reagent (Thermo Fisher Scientific). The cells were incubated with the transfection mix for 5–6 hours in DMEM + Glutamax (Thermo Fisher Scientific)/10% FBS Superior (Biochrom) without antibiotics. The cells were then incubated in DMEM + Glutamax/10% FBS/1% penicillin-streptomycin (PenStrep, Sigma) overnight. After 24 hours, they were rinsed in PBS and lysed with PLB buffer for Luciferase Assay (Promega). The supernatants were used to perform the luciferase assay, using the Dual-Luciferase Reporter Assay System (Promega), following the manufac-turer's instructions. Each experiment was carried out in triplicates (three wells per condition) in four independent experiments.

## Cell line

We used human Embryonic Kidney cells (HEK293T, ATCC Cat# CRL-3216), which were certified by STR profiling by ATCC, and tested negative for mycoplasma contamination.

## RNA-seq

48 hpf *Tg(myl7:BFP-CAAX) snai1b*$^{+/+}$ and *snai1b*$^{-/-}$ hearts were manually dissected using forceps. Approximately 20 hearts per replicate were pooled, and total RNA was isolated using the miRNeasy micro kit, combined with on-column DNase digestion. Total RNA and library integrity were verified with LabChip Gx Touch 24 (Perkin Elmer). Approximately 10 ng of total RNA was used as input for SMART-Seq v4 Ultra Low Input RNA Kit (Takara Clontech) for cDNA pre-amplification. Obtained full-length cDNA was checked on LabChip GX Touch 24 and fragmented by Ultrasonication by E220 machine (Covaris). Final Library Preparation was performed by Low Input Library Prep Kit v2 (Takara Clontech). Sequencing was performed on a NextSeq500 instrument (Illumina) using v2 chemistry, resulting in an average of 30M reads per library with 1 × 75 bp single-end setup. The resulting raw reads were assessed for quality, adapter content, and duplication rates with FastQC (available online at http://www.bioinformatics.babraham.ac.uk/projects/fastqc). Trimmomatic version 0.39 was used to trim reads with a quality drop below a mean of Q20 in a window of 10 nucleotides (Bolger et al., Trimmomatic: a flexible trimmer for Illumina sequence data). Only reads between 30 and 150 nucleo-tides were used in subsequent analyses. Trimmed and filtered reads were aligned versus the Ensembl Zebrafish genome version DanRer11 (GRCz11.92) using STAR 2.6.1d with the parameter 'outFilterMismatchNoverLmax 0.1' to increase the maximum ratio of mismatches to mapped length to 10% (*Dobin et al., 2013*). The number of reads aligning to genes was counted with feature Counts 1.6.5 tool from the Subread package (*Liao et al., 2014*). Only reads mapping at least par-tially inside exons were admitted and aggregated per gene, while reads overlapping multiple genes or aligning to multiple regions were excluded from further analyses. Differentially expressed genes were identified using DESeq2 version 1.18.1 (*Love et al., 2014*). The Ensembl annotation was enriched with UniProt data (release 06.06.2014) based on Ensembl gene identifiers (Activities at the Universal Protein Resource (UniProt)).

For the gene ontology analysis, all genes with a p-value≤0.05 were used as a query list. Genes with >5 normalized reads in at least one sample were used as a background list. The analysis was performed with the Gitools 2.3.1 (http://www.gitools.org) software. Z-scores were calculated using the default settings, and multiple test correction with Benjamini–Hochberg FDR was performed.

## Transmission electron microscopy (TEM)

Larvae were collected at 60 hpf from a wild-type or mutant incross. The embryos were immediately fixed in ice-cold 1% PFA, 2% glutaraldehyde in 0.1 M sodium cacodylate buffer (pH 7.4) for 30 min

on ice, and then stored at 4°C overnight. Samples were washed in 0.1 M sodium cacodylate buffer and postfixed in 2% (w/v) $OsO_4$, followed by *en bloc* staining with 2% uranyl acetate. Samples were dehydrated with a graded series of washes in acetone, transferred to acetone/Epon solutions, and eventually embedded in Epon. Ultra-thin sections (approximately 70 nm thick) obtained with a Reichert-Jung Ultracut E microtome were collected on copper slot grids. Sections were post-stained with 2% uranyl acetate for 20 min and 1% lead citrate for 2 min. Sections were examined with a Jeol JEM-1400 Plus transmission electron microscope (Jeol, Japan), operated at an accelerating voltage of 120 kV. Digital images were recorded with an EM-14800 Ruby Digital CCD camera unit (3296px × 2472px).

### Randomization and blinding procedures

All experiments using *snai1b* mutants were randomized as follows: animals from heterozygous crosses were collected, imaged, and analysed, and subsequently genotyped. For all immunostainings, the genotyping was performed after the analysis. The only exceptions were for the RNAseq, TEM, and *tnnt2a* MO experiments, for which the mutants were obtained from maternal zygotic incrosses using *snai1b* zygotic mutants (approximately 70% of them reach adulthood). All experiments shown in *Figure 1—figure supplement 4* were performed with first generation cousin animals. Transgenic animals were selected by fluorescence before imaging, and therefore could not be randomized. The investigators were blinded to allocation during experiments and outcome assessment whenever possible.

### Statistical analysis

All statistical analyses were performed in GraphPad Prism (version 6.07). A Gaussian distribution was tested for every sample group using the D'Agostino–Pearson omnibus normality test. For the experiments that passed the normality test, all samples were further analysed using the following parametric tests: the Student's t-test for comparison of two samples or the one-way ANOVA test followed by correction for multiple comparisons with Dunn's test for three or more samples. For all the experiments that did not pass the normality test, all samples were further analysed using non-parametric tests: p-values were determined using the Mann–Whitney test for comparison of two samples or the Kruskal–Wallis test followed by correction for multiple comparisons with Dunn's test for three or more samples.

## Acknowledgements

This work was supported by funds from the Max Planck Society to DYRS, a European Molecular Biology Organization (EMBO) Advanced Fellowship (ALTF 642-2018) and a Canadian Institute for Health Research Fellowship (293898) to FG, and an EMBO fellowship (LTF 1569-2016), a Humboldt fellowship and a Cardio-Pulmonary Institute Grant (EXC 2026, project ID 390649896) to RP. We would like to thank Michelle Collins, Paolo Panza, Chi-Chung Wu, Mridula Balakrishnan, Srinivas Allanki, Giulia Boezio, Simon Perathoner, and Honorine Destain for comments on the manuscript, Prof. Akira Nagafuchi for the α-catenin epitope α-18 antibody, and Gabrielius Jakutis for help with the HEK293T cells.

## Additional information

### Competing interests

Didier YR Stainer: Senior editor, *eLife*. The other authors declare that no competing interests exist.

### Funding

| Funder | Grant reference number | Author |
| --- | --- | --- |
| Max Planck Society | | Alessandra Gentile<br>Anabela Bensimon-Brito<br>Rashmi Priya<br>Hans-Martin Maischein<br>Janett Piesker |

| | | Stefan Guenther |
| | | Felix Gunawan |
| | | Didier YR Stainier |
| Deutsches Zentrum für Herz-Kreislaufforschung | | Rashmi Priya |
| | | Stefan Guenther |
| | | Felix Gunawan |
| | | Didier YR Stainier |
| European Molecular Biology Organization | ALTF 642–2018 | Felix Gunawan |
| Cardio Pulmonary Institute Grant | EXC 2026 390649896 | Rashmi Priya |
| CIHR | 293898 | Felix Gunawan |
| European Molecular Biology Organization | LTF 1569-2016 | Rashmi Priya |

The funders had no role in study design, data collection and interpretation, or the decision to submit the work for publication.

### Author contributions

Alessandra Gentile, Conceptualization, Formal analysis, Validation, Investigation, Visualization, Methodology, Writing - original draft, Writing - review and editing; Anabela Bensimon-Brito, Conceptualization, Supervision, Methodology, Writing - review and editing; Rashmi Priya, Conceptualization, Writing - review and editing; Hans-Martin Maischein, Methodology; Janett Piesker, Investigation, Methodology, Writing - review and editing; Stefan Guenther, Formal analysis, Investigation, Methodology, Writing - review and editing; Felix Gunawan, Conceptualization, Supervision, Methodology, Writing - original draft, Writing - review and editing; Didier YR Stainier, Conceptualization, Supervision, Funding acquisition, Writing - original draft, Project administration, Writing - review and editing

### Author ORCIDs

Alessandra Gentile  https://orcid.org/0000-0002-5423-7295
Anabela Bensimon-Brito  http://orcid.org/0000-0003-1663-2232
Rashmi Priya  https://orcid.org/0000-0002-0510-7515
Felix Gunawan  https://orcid.org/0000-0002-5592-9680
Didier YR Stainier  https://orcid.org/0000-0002-0382-0026

### Ethics

Animal experimentation: Zebrafish husbandry was performed in accordance with institutional (MPG) and national (German) ethical and animal welfare regulation. All procedures performed on animals conform to the guidelines from Directive 2010/63/EU of the European Parliament on the protection of animals used for scientific purposes and were approved by the Animal Protection Committee (Tierschutzkommission) of the Regierungspräsidium Darmstadt (reference: B2/1218).

### Decision letter and Author response

Decision letter https://doi.org/10.7554/eLife.66143.sa1
Author response https://doi.org/10.7554/eLife.66143.sa2

## Additional files

### Supplementary files

- Supplementary file 1. Table of primers.
- Supplementary file 2. Table of Ct values from RT-qPCR experiments.
- Supplementary file 3. Luciferase assay raw values.
- Transparent reporting form

## Data availability

Sequencing data have been deposited in GEO under accession code GSE162604.

The following dataset was generated:

| Author(s) | Year | Dataset title | Dataset URL | Database and Identifier |
|---|---|---|---|---|
| Gentile A, Guenther S | 2020 | RNAseq of snai1b mutant hearts | https://www.ncbi.nlm.nih.gov/geo/query/acc.cgi?acc=GSE162604 | NCBI Gene Expression Omnibus, GSE162604 |

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
