## [Decision Letter]

**Acceptance summary:**

The examination into how EMT related transcriptional pathways regulate cardiomyocyte gene expression provides critical insight into how heart wall integrity is developed and maintained. These current findings will also likely have an important impact on how these pathways regulate the injury response to the heart.

**Decision letter after peer review:**

Thank you for submitting your article "The EMT transcription factor Snai1 maintains myocardial wall integrity by repressing intermediate filament genes" for consideration by *eLife*. Your article has been reviewed by 3 peer reviewers, and the evaluation has been overseen by Edward Morrisey as the Senior and Reviewing Editor. The following individuals involved in review of your submission have agreed to reveal their identity: Chinmay M Trivedi (Reviewer #1); Benoit Bruneau (Reviewer #3).

Essential revisions:

1. Additional immunohistochemistry to assess the shape changes in Snai1 deficient cardiomyocytes.

2. Further examination of the effects of Desmin over expression on cardiomyocyte behavior as noted by reviewer 2 and 3.

3. All noted text changes from the reviewers should be addressed through careful editing.

*Reviewer #1 (Recommendations for the authors):*

Gentile A et al. show a novel role of Snai1b in growth regulation of zebrafish myocardial wall. Specifically, authors show that zebrafish lacking Snai1b exhibit cardiac looping defects (~50% penetrance), consistent with previously described morpholino mediated Snai1b knockdown phenotype. Extruding cardiomyocytes away from cardiac lumen, mostly in the atrioventricular canal region were observed in remaining 50% of Snai1b knockout zebrafish. Using RNA-seq, authors identified several dysregulated genes, including enrichment of intermediate filament genes in Snai1b knockout zebrafish. Among these dysregulated genes, authors suggest that increased Desmin expression and its aberrant localization promote cardiomyocyte extrusion in Snai1b knockout zebrafish hearts. Overall, present manuscript describes a novel phenomenon during cardiac development, hence, it is of interest to developmental biologists. Major concerns are:

1. Snai1 is known to affect cushion formation in atrioventricular canal region. It would be helpful to establish cause and effect relationship for Snai1b in this region. Zebrafish lack global Snai1b expression – so it would be helpful to show if defective cushion promotes cardiomyocyte extrusion in atrioventricular canal region. Tnnt2 morpholino experiments provides some insights, however, it does not rule out role of defective atrioventricular cushion (defective EMT).

2. For Figure 2 – additional histology / immunohistology to show extrusion, cohesion, and orientation of cardiomyocytes at a section level (2D) in Snai1b knockout hearts could help to characterize phenotype at a cellular level. It is assumed that all cardiomyocytes lack Snai1b protein (immunostaining would help), however, only few cardiomyocyte show extrusion. Minor point – Cartoon images in figure 2 are somewhat disconnected from immunostaining images.

3. Do Snai1b knockout hearts exhibit defective contractile phenotype? Is there a cardiac phenotype in surviving adult zebrafish? Do RNA-seq and SEM from adult zebrafish heart represent embryonic extrusion and intermediate filament defects?

4. It is unclear why only few cardiomyocytes show extrusion when most of cardiomyocytes, if not all, overexpressing Desmin gene.

5. Molecular link connecting Snai1b and cardiac filaments genes is not determined.

*Reviewer #2 (Recommendations for the authors):*

An intact myocardium is essential for cardiac function, yet much remains unknown regarding the cell biological mechanisms maintaining this specialized epithelium during embryogenesis. In this manuscript, Gentile and colleagues discover a novel role for the repressive transcription factor Snai1b in supporting myocardial integrity. In the absence of Snai1b, cardiomyocytes exhibit an enrichment of intermediate filament genes, including desmin b. In addition, the authors detect mislocalization of Desmin, along with adherens junction and actomyosin components, to the basal membrane in snai1b mutant cardiomyocytes, and these mutant cells exhibit an increased likelihood of extrusion from the myocardium. Ultimately, the authors put forward a model wherein Snai1b protects cardiomyocytes from extrusion at least in part by regulating the amount and organization of Desmin in the cell, thereby supporting myocardial integrity.

Overall, the authors highlight an important aspect of epithelial maintenance in an environment that experiences significant biomechanical stress due to cardiac function. By generating a promoter-less allele of snai1b, the authors have created a clean genetic model in which to work. Coupled with beautiful microscopy and transcriptomics, this story has the potential to enlighten both cell biologists and cardiovascular biologists on the underpinnings of myocardial integrity. However, clarifications regarding the overall model would be particularly beneficial for the reader.

1. A clearer discussion of the proposed molecular mechanism for Snai1b function would aid a reader's overall contextualization of this work. At one point, the authors suggest that Snai1b regulates N-cadherin localization to adherens junctions, thereby stabilizing actomyosin tension at cell junctions. Later, it is suggested that Desmin activates the actomyosin contractile network at the basal membrane. It is unclear whether the authors believe that these are separate events or whether they may be coupled, perhaps through Desmin disruption at the lateral membranes, leading to modifications in nearby adherens junctions. A more thorough investigation of the phenotype resulting from desmin b overexpression may clarify this relationship.

2. It appears that extruded cells do not bud off from the myocardium, but rather remain on the apical surface of the existing myocardium. However, it is unclear whether this change in tissue architecture affects cardiac function or the overall morphology of the chamber. A brief discussion of these possibilities would have helped to contextualize the significance of this phenotype.

3. The authors show that cardiomyocyte extrusion is most prevalent near the atrioventricular canal, and they suggest that this regionalized effect is due to the different types of extrinsic factors, like biomechanical forces, that this region experiences. However, it is also possible that regional differences in certain intrinsic factors are involved, such as junctional plasticity, actomyosin activity at the basal membrane, etc. To distinguish between these possibilities, it would have been informative to know whether the extent of N-cadherin/α-18/p-Myosin/Desmin mislocalization varies depending on the regional location of cardiomyocytes within the snai1b mutant heart. For example, do cardiomyocytes near the atrioventricular canal exhibit more extreme effects on N-cadherin/α-18/p-Myosin/Desmin localization than cardiomyocytes in further away portions of the ventricle? Or, do these cells exhibit similar degrees of protein mislocalization, but cells near the atrioventricular canal have a lower threshold for extrusion?

The following adjustments and/or clarifications would strengthen the manuscript.

1. Regarding the precise mechanism by which extrusion occurs, it would be helpful to know which aspects are a direct cause of Desmin overexpression/mislocalization. The authors begin to investigate this question by staining for α-18. However, these data seem incomplete, as they are not subdivided into extruding vs. non-extruding cells (as in other experiments). Additionally, if the authors wish to assemble further support for the model that Desmin regulates actomyosin activity at the basal surface, then assaying p-Myosin would bolster this idea, rather than relying on α-18 as an indicator of contractility. Finally, assaying N-cadherin localization in Desmin overexpressing cells would enable the authors to distinguish whether the N-cadherin mislocalization observed in snai1b mutants is dependent on Desmin overexpression or a parallel effect of Snai1b deficiency – both of these would be interesting outcomes that would help to clarify their overall model.

2. Given the reduction of N-cadherin on lateral membranes and the proposal that junctional actomyosin is destabilized in snai1b mutant cardiomyocytes, one might expect that adherens junctions are not intact in mutants. The authors did not compare the integrity and/or abundance of adherens junctions in their EM experiments, which seems like a missed opportunity. Are adherens junctions visible in any of the existing EM data?

3. In the N-cadherin fluorescence intensity profiles, why is the signal routinely higher on one side the of the cell? Shouldn't the signal be equivalent, independent of the side where the profile begins?

4. Is cardiac function affected in snai1b mutants? If they are functionally deficient (e.g. reduced contractility), this would somewhat lessen the impact of the mutant-into-wild-type transplant experiments, as it has been observed that cardiomyocytes with impaired sarcomeric contraction are extruded when they are mosaically present within a functional myocardium. If mutants are functionally normal, however, adding this information would strengthen the meaningfulness of the cell autonomy experiments.

5. Do the authors believe that the cardiomyocytes that are not extruded in snai1b mutants are affected in any meaningful way? For example, is the observed trabeculation defect believed to be related to the extrusion phenotype?

*Reviewer #3 (Recommendations for the authors):*

Gentile, A. et al. generated snai1b mutant zebrafish embryos and showed that loss of Snai1b led to two mutant phenotypes in the heart: i) hearts with clear looping defects, ii) hearts without looping defects that displayed abnormal cardiomyocyte (CM) extrusion. The authors focused on the second class of mutants and found that loss of Snai1b led to reduction of N-cadherin at cell junctions and basal accumulation of phosphorylated myosin light chain and the α-18 epitope of α-catenin, indicative of mechanical activation. Bulk RNA-sequencing of isolated hearts revealed an upregulation of intermediate filament (IF) genes in Snai1b mutants, and of particular interest, the authors identified upregulation of the muscle-specific IF gene desmin b. Immunofluorescent imaging revealed that Desmin was not only upregulated in Snai1b mutants, but mis-localized away from cell junctions and accumulated at the basal side of extruding cells along with actomyosin machinery. Accordingly, CM-specific overexpression of Desmin was sufficient to promote cell extrusion.

The presented work is particularly interesting because it identifies a new role for the Snai1b transcription factor in maintaining proper tissue structure, independent of its typical function in regulating epithelial to mesenchymal transition (EMT). Overall, the experiments were well designed and controlled, and the data is clearly and logically presented. However, some of the findings could be explained by alternative hypotheses and other interesting aspects of the data were left unexplored.

One hypothesis that was not sufficiently discussed is that loss of Snai1b may prevent cardiomyocytes from undergoing the EMT that is necessary for normal delamination and trabeculation, and thus cells are instead extruded away from the lumen to prevent overcrowding in the developing myocardium. In fact, the authors present evidence that EMT is blocked and acknowledge that extrusion is a known mechanism for preventing overcrowding. It would be interesting to see whether extrusion away from the lumen also occurs if EMT is blocked through other means.

The authors show that extruding cells do not seem to be dead or dying, and that a small number of CMs do extrude in wild type embryos. This raises the intriguing possibility that some amount of CM extrusion is necessary for normal development and that these cells may give rise to epicardial or other cell types. Live-imaging and lineage-tracing studies would inform whether the extrusion observed in mutant embryos is an enhancement of a normal morphogenetic process or an additional abnormal response to loss of Snai1 function.

One particularly interesting observation that was left unexplored was the identification of a second class of Snai1b mutants with defective heart looping. It isn't clear whether these embryos also display enhanced CM extrusion, or if there are other clearly aberrant cell behaviors. Furthermore, it would be very interesting to know whether there is any evidence that the defective looping is due to the same changes in cytoskeletal gene expression and protein organization observed in the class of Snai1b mutants that were detailed throughout the manuscript.

The authors suggest that Snai1b regulates Desmin in two ways: 1) overall expression levels, and 2) post-translationally to control its localization at cell junctions. Although the first claim is sufficiently supported, the second claim lacks experimental evidence. An alternative explanation is that overexpression of Desmin in response to loss of Snai1b leads to mislocalization independent of an interaction with Snai1b. This point could be clarified by examining Desmin localization in the desmb overexpression system. In addition, assaying for co-IP of Snai1b and Desmin could demonstrate a direct interaction between the two and better support a role for Snai1 in regulating post-translational localization of Desmin.

Although the authors convincingly show that Desmin accumulates with other contractile machinery at the basal side of extruding CMs in Snai1b muntants, additional evidence is needed to support a causal link between basal Desmin accumulation and extrusion. For instance, if knockdown or inhibition of Desmin prevents extrusion in the Snai1b mutants, the causal relationship would be much clearer.

---

## [Author Response]

Essential revisions:1. Additional immunohistochemistry to assess the shape changes in Snai1 deficient cardiomyocytes.

We have now included additional in-depth characterization of the shape changes in *snai1b* mutant CMs. We quantified CM surface areas and aspect ratios, as well as ventricular volumes, at 52 and 74 hpf. These quantifications were carried out in live fish carrying the transgenic CM membrane marker *Tg(myl7:HRAS-EGFP)* in order to preserve CM morphology and avoid potential damage from the immunostaining procedure. We found that the *snai1b* mutant CMs are significantly smaller and more rounded at both time points. This result suggests that the reduction in intercellular adhesion and changes in cytoskeletal architecture caused by the loss of Snai1b lead to morphological changes and abnormal rounding of the CMs. We also found significantly smaller ventricular volumes at 52 and 74 hpf in *snai1b* mutant hearts, suggesting that the cellular defects impact tissue architecture. These new results are shown in Figure 1 – supplement 5.

2. Further examination of the effects of Desmin over expression on cardiomyocyte behavior as noted by reviewer 2 and 3.

We showed in the previous version of the manuscript that *desmb* overexpression in CMs leads to basal localization of the α-catenin epitope α-18 and cell extrusion. We have now included additional data using anti-Desmin, anti-p-myosin, and anti-N-cadherin immunostaining to further characterize the effects of *desmb* overexpression in CMs. First, we observed the enrichment of Desmin at the basal domain of *desmb* overexpressing CMs. We also observed in *desmb* overexpressing CMs an increased localization of p-myosin in their basal domain (similar to the basal enrichment of the α-catenin epitope α-18 shown in figure 4D-D’) and a reduction of junctional N-cadherin. These results further show that CM-specific *desmb* overexpression leads to dysregulation of the actomyosin cytoskeleton and junctional adhesion, phenocopying *snai1b* mutant CMs. We have also included more detailed quantitative analysis to differentiate the fluorescence signal between extruding and non-extruding *desmb* overexpressing CMs. These new results are shown in Figure 4 and Figure 4 – supplement 1.

3. All noted text changes from the reviewers should be addressed through careful editing.

We have carefully edited the text as suggested by the Reviewers.

Reviewer #1 (Recommendations for the authors):[…] Major concerns are:1. Snai1 is known to affect cushion formation in atrioventricular canal region. It would be helpful to establish cause and effect relationship for Snai1b in this region. Zebrafish lack global Snai1b expression – so it would be helpful to show if defective cushion promotes cardiomyocyte extrusion in atrioventricular canal region. Tnnt2 morpholino experiments provides some insights, however, it does not rule out role of defective atrioventricular cushion (defective EMT).

We thank the reviewer for these suggestions. While we agree that this point is interesting, it must be noted that atrioventricular (AV) valve formation in zebrafish starts at ⁓56 hpf, with the collective migration of the valve endothelial cells (Gunawan et al., 2019; Gunawan et al., 2020). Zebrafish heart valves are functional starting at approximately 72 hpf, when they can efficiently close the lumen (Gunawan et al., 2019; Gunawan et al., 2020). Thus, the endothelial-to-mesenchymal transition process in the AV canal takes place after CMs start extruding in *snai1b* mutants (48 hpf).

Nevertheless, we examined valve formation in *snai1b* mutants and observed that the early stages of valve development seem unaffected, and that wild-type like valve leaflets appear by 72 hpf (Author response image 1). Together, these data indicate that the CM extrusion defects in *snai1b* mutants are not a secondary effect of valve dysfunction.

**Author response image 1. sa2fig1:** Single-plane images of *Tg(kdrl:nls-mCherry)snai1b^+/+^* (A) and *snai1b^-/-^* (B) valve leaflets at 72 hpf. (*snai1b^+/+^,* n=9; *snai1b^-/-^,* n=11). Scale bars: 20 µm. n, number of embryos.

2. For Figure 2 – additional histology / immunohistology to show extrusion, cohesion, and orientation of cardiomyocytes at a section level (2D) in Snai1b knockout hearts could help to characterize phenotype at a cellular level. It is assumed that all cardiomyocytes lack Snai1b protein (immunostaining would help), however, only few cardiomyocyte show extrusion. Minor point – Cartoon images in figure 2 are somewhat disconnected from immunostaining images.

We thank the reviewer for these suggestions. While we agree with the point regarding the Snai1b immunostaining, there are no commercially available or published antibodies that detect zebrafish Snai1b, particularly one that differentiates between Snai1a and Snai1b. To better characterize the *snai1b* mutant phenotype at the cellular level, we have now included quantification of CM apical surface areas and aspect ratios at 52 and 74 hpf. We found that the CMs in *snai1b* mutants appear smaller and more rounded compared with those in wild-type embryos. We also found a smaller ventricular volume in *snai1b* mutant hearts, potentially due to the changes in CM shape. These new results are shown in Figure 1 – supplement 5. We also changed the cartoons in Figure 2, as suggested.

3. Do Snai1b knockout hearts exhibit defective contractile phenotype? Is there a cardiac phenotype in surviving adult zebrafish? Do RNA-seq and SEM from adult zebrafish heart represent embryonic extrusion and intermediate filament defects?

We thank the reviewer for these comments. However, characterizing the *snai1b* mutant adult phenotypes is beyond the scope of this manuscript. It is important to clarify that the RNA-seq and SEM experiments were performed in embryonic hearts.

4. It is unclear why only few cardiomyocytes show extrusion when most of cardiomyocytes, if not all, overexpressing Desmin gene.

We agree with the reviewer. As we showed by immunostaining in wild-type hearts (Figure 3I), a subset of CMs – the few extruding wild-type CMs – exhibit Desmin enrichment in their basal domain. We speculate that only this subset of CMs exhibits basal Desmin enrichment because their position within the myocardium exposes them to higher mechanical forces due to increased blood flow and looping morphogenesis, which in turn raises their propensity to extrude. Indeed, as we show in Figure 1 – supplement 1F, most of the CM extrusions in *snai1b* mutants are observed at the AV canal, where CMs experience the highest level of mechanical forces (Lombardo et al., 2019; Campinho et al., 2020).

5. Molecular link connecting Snai1b and cardiac filaments genes is not determined.

We have now used a luciferase assay in HEK293T cells to test the regulation of *desmb* expression by Snai1b. It was previously shown by ChIP-seq in mouse skeletal myoblasts that Snai1 can bind to the proximal promoter of *Desmin* (Soleimani et al., 2012). Our *in silico* analysis uncovered an 800 base pair region upstream of the start codon of zebrafish *desmb* that exhibits a high degree of similarity (>45%) with the mammalian sequence and is thus a promising proximal promoter for *desmb*. Furthermore, Kürekçi et al. recently reported that the zebrafish *desmb* promoter contains putative Snai1b-binding sites (Kayman Kürekçi et al., 2021). We cloned this 800 bp region upstream of a luciferase reporter and co-transfected the resulting plasmid with a plasmid expressing zebrafish Snai1b, which led to a significant decrease of luciferase activity compared with the proximal promoter alone. These data suggest that Snai1b binds to the proximal promoter of *desmb* and represses its transcription, potentially implicating Snai1b as a direct regulator of *desmb* expression. This new result is shown in Figure 3 – supplement 1D.

Reviewer #2 (Recommendations for the authors):An intact myocardium is essential for cardiac function, yet much remains unknown regarding the cell biological mechanisms maintaining this specialized epithelium during embryogenesis. In this manuscript, Gentile and colleagues discover a novel role for the repressive transcription factor Snai1b in supporting myocardial integrity. In the absence of Snai1b, cardiomyocytes exhibit an enrichment of intermediate filament genes, including desmin b. In addition, the authors detect mislocalization of Desmin, along with adherens junction and actomyosin components, to the basal membrane in snai1b mutant cardiomyocytes, and these mutant cells exhibit an increased likelihood of extrusion from the myocardium. Ultimately, the authors put forward a model wherein Snai1b protects cardiomyocytes from extrusion at least in part by regulating the amount and organization of Desmin in the cell, thereby supporting myocardial integrity.Overall, the authors highlight an important aspect of epithelial maintenance in an environment that experiences significant biomechanical stress due to cardiac function. By generating a promoter-less allele of snai1b, the authors have created a clean genetic model in which to work. Coupled with beautiful microscopy and transcriptomics, this story has the potential to enlighten both cell biologists and cardiovascular biologists on the underpinnings of myocardial integrity. However, clarifications regarding the overall model would be particularly beneficial for the reader.1) A clearer discussion of the proposed molecular mechanism for Snai1b function would aid a reader's overall contextualization of this work. At one point, the authors suggest that Snai1b regulates N-cadherin localization to adherens junctions, thereby stabilizing actomyosin tension at cell junctions. Later, it is suggested that Desmin activates the actomyosin contractile network at the basal membrane. It is unclear whether the authors believe that these are separate events or whether they may be coupled, perhaps through Desmin disruption at the lateral membranes, leading to modifications in nearby adherens junctions. A more thorough investigation of the phenotype resulting from desmin b overexpression may clarify this relationship.

We thank the reviewer for these comments. We have now included Desmin and p-myosin immunostaining of *desmb* overexpressing CMs. We observed an increased level of Desmin protein in the *desmb* overexpressing CMs, as well as its basal localization. We also observed increased p-myosin localization basally, as we did in *snai1b* mutant CMs. N-cadherin immunostaining at the junctions was reduced in *desmb* overexpressing CMs, as in *snai1b* mutant CMs. Altogether, these results indicate that myocardial-specific *desmb* overexpression phenocopies *snai1b* mutants. We have also included deeper quantitative analysis of the immunostaining, now distinguishing the results between extruding and non-extruding *desmb* overexpressing CMs. These new results are shown in Figure 4F-I’ and Figure 4 – supplement 2.

Although both reduced junctional N-cadherin and abnormal basal localization of actomyosin factors are consistently observed in *snai1b* mutants and in *desmb* overexpressing embryos, additional tools will need to be developed and used to determine whether they are separate or coupled events.

2) It appears that extruded cells do not bud off from the myocardium, but rather remain on the apical surface of the existing myocardium. However, it is unclear whether this change in tissue architecture affects cardiac function or the overall morphology of the chamber. A brief discussion of these possibilities would have helped to contextualize the significance of this phenotype.

We thank the reviewer for this interesting point. We have now included a time-lapse spinning disk movie of wild-type and *snai1b* mutant hearts from 52 to 70 hpf. At the starting timepoint (t_0_) in *snai1b* mutant hearts, we observed extruding CMs that were still embedded within the myocardium. Within 6 hours, we did not observe extruding CMs in the same location as we had at t_0_, but instead found CMs outside of the myocardial wall and they remained in the pericardial cavity for several hours. These new results suggest that CMs do indeed extrude out of the myocardium in *snai1b* mutant hearts, and they are shown in Figure 1 – supplement 1I-K and video 1.

Additionally, we quantified the heart rate, ejection fraction, and fractional shortening at 52 and 74 hpf. At 52 hpf, we did not find significant differences between wild-type and *snai1b* mutants, but at 74 hpf, the heart rate, ejection fraction, and fractional shortening were significantly lower in *snai1b* mutants compared to wild types. Furthermore, *snai1b* mutants exhibited reduced ventricular volume at 52 and 74 hpf. As the reduction in cardiac function occurs after CMs start to extrude, these data indicate that CM extrusion has an impact on the overall morphology of the ventricle and cardiac function. These new results are shown in Figure 1 – supplement 5.

3) The authors show that cardiomyocyte extrusion is most prevalent near the atrioventricular canal, and they suggest that this regionalized effect is due to the different types of extrinsic factors, like biomechanical forces, that this region experiences. However, it is also possible that regional differences in certain intrinsic factors are involved, such as junctional plasticity, actomyosin activity at the basal membrane, etc. To distinguish between these possibilities, it would have been informative to know whether the extent of N-cadherin/α-18/p-Myosin/Desmin mislocalization varies depending on the regional location of cardiomyocytes within the snai1b mutant heart. For example, do cardiomyocytes near the atrioventricular canal exhibit more extreme effects on N-cadherin/α-18/p-Myosin/Desmin localization than cardiomyocytes in further away portions of the ventricle? Or, do these cells exhibit similar degrees of protein mislocalization, but cells near the atrioventricular canal have a lower threshold for extrusion?

We thank the reviewer for this interesting point. We hypothesize that CMs closer to the AV canal exhibit more severe effects on N-cadherin/α-catenin epitope α-18/p-myosin/Desmin localization, due to the higher mechanical forces they experience (Lombardo et al., 2019; Campinho et al., 2020). However, our immunostaining procedure for zebrafish embryos requires deyolking to allow access of the tissue to the antibody, which unfortunately leads to the loss of the atrium and the part of the AV canal closest to the atrium; thus, this procedure renders it difficult to perform quantitative analysis of the immunostaining signal throughout the heart.

The following adjustments and/or clarifications would strengthen the manuscript.1. Regarding the precise mechanism by which extrusion occurs, it would be helpful to know which aspects are a direct cause of Desmin overexpression/mislocalization. The authors begin to investigate this question by staining for α-18. However, these data seem incomplete, as they are not subdivided into extruding vs. non-extruding cells (as in other experiments). Additionally, if the authors wish to assemble further support for the model that Desmin regulates actomyosin activity at the basal surface, then assaying p-Myosin would bolster this idea, rather than relying on α-18 as an indicator of contractility. Finally, assaying N-cadherin localization in Desmin overexpressing cells would enable the authors to distinguish whether the N-cadherin mislocalization observed in snai1b mutants is dependent on Desmin overexpression or a parallel effect of Snai1b deficiency – both of these would be interesting outcomes that would help to clarify their overall model.

We thank the reviewer for these comments. We have now included Desmin and p-myosin immunostaining of *desmb* overexpressing CMs. We observed an increased level of Desmin protein, as well as its basal localization. We also observed increased p-myosin localization basally, as we did in *snai1b* mutant CMs. N-cadherin immunostaining at the junctions were reduced in *desmb* overexpressing CMs, as in *snai1b* mutant CMs. Altogether, these results indicate that myocardial-specific *desmb* overexpression phenocopies *snai1b* mutants. We have also included deeper quantitative analyses of the immunostaining, now distinguishing the results between extruding and non-extruding *desmb* overexpressing CMs. These new results are shown in Figure 4F-I’ and Figure 4 – supplement 2.

2. Given the reduction of N-cadherin on lateral membranes and the proposal that junctional actomyosin is destabilized in snai1b mutant cardiomyocytes, one might expect that adherens junctions are not intact in mutants. The authors did not compare the integrity and/or abundance of adherens junctions in their EM experiments, which seems like a missed opportunity. Are adherens junctions visible in any of the existing EM data?

We thank the reviewer for this interesting comment. Unfortunately, adherens junctions were not visible in any of the existing EM data.

3. In the N-cadherin fluorescence intensity profiles, why is the signal routinely higher on one side the of the cell? Shouldn't the signal be equivalent, independent of the side where the profile begins?

We have now fixed these profiles and thank the reviewer for pointing out this issue.

4. Is cardiac function affected in snai1b mutants? If they are functionally deficient (e.g. reduced contractility), this would somewhat lessen the impact of the mutant-into-wild-type transplant experiments, as it has been observed that cardiomyocytes with impaired sarcomeric contraction are extruded when they are mosaically present within a functional myocardium. If mutants are functionally normal, however, adding this information would strengthen the meaningfulness of the cell autonomy experiments.

We thank the reviewer for this important point. We compared cardiac function in *snai1b^+/+^* and *snai1b^-/-^* embryos by quantifying their heart rate, ejection fraction, and fractional shortening at 52 hpf, a timepoint at which CM extrusions are already observed in *snai1b* mutants. Notably, we found that at this stage loss of *snai1b* does not significantly affect cardiac function (Figure 1—figure supplement 4).

5. Do the authors believe that the cardiomyocytes that are not extruded in snai1b mutants are affected in any meaningful way? For example, is the observed trabeculation defect believed to be related to the extrusion phenotype?

We thank the reviewer for these questions. We hypothesize that the transcriptome of all CMs in *snai1b* mutants are affected in a similar manner. However, the extruding CMs are potentially exposed to higher mechanical forces from being in, or close to, the AV canal. The trabeculation defects in *snai1b* mutants could indeed be related to the CM extrusion phenotype as trabeculation requires the actomyosin machinery to be enriched in the apical domain to induce apical constriction and basal delamination (Priya et al., 2020). Thus, the ectopic enrichment of actomyosin factors in the basal domain of CMs in *snai1b* mutants might interfere with trabeculation.

Reviewer #3 (Recommendations for the authors):Gentile, A. et al. generated snai1b mutant zebrafish embryos and showed that loss of Snai1b led to two mutant phenotypes in the heart: i) hearts with clear looping defects, ii) hearts without looping defects that displayed abnormal cardiomyocyte (CM) extrusion. The authors focused on the second class of mutants and found that loss of Snai1b led to reduction of N-cadherin at cell junctions and basal accumulation of phosphorylated myosin light chain and the α-18 epitope of α-catenin, indicative of mechanical activation. Bulk RNA-sequencing of isolated hearts revealed an upregulation of intermediate filament (IF) genes in Snai1b mutants, and of particular interest, the authors identified upregulation of the muscle-specific IF gene desmin b. Immunofluorescent imaging revealed that Desmin was not only upregulated in Snai1b mutants, but mis-localized away from cell junctions and accumulated at the basal side of extruding cells along with actomyosin machinery. Accordingly, CM-specific overexpression of Desmin was sufficient to promote cell extrusion.The presented work is particularly interesting because it identifies a new role for the Snai1b transcription factor in maintaining proper tissue structure, independent of its typical function in regulating epithelial to mesenchymal transition (EMT). Overall, the experiments were well designed and controlled, and the data is clearly and logically presented. However, some of the findings could be explained by alternative hypotheses and other interesting aspects of the data were left unexplored.One hypothesis that was not sufficiently discussed is that loss of Snai1b may prevent cardiomyocytes from undergoing the EMT that is necessary for normal delamination and trabeculation, and thus cells are instead extruded away from the lumen to prevent overcrowding in the developing myocardium. In fact, the authors present evidence that EMT is blocked and acknowledge that extrusion is a known mechanism for preventing overcrowding. It would be interesting to see whether extrusion away from the lumen also occurs if EMT is blocked through other means.

We thank the reviewer for these interesting questions regarding CM overcrowding, EMT, and CM extrusion. To test the hypothesis that CMs in *snai1b* mutants are extruding to prevent overcrowding in the developing myocardium, we treated embryos with an ErbB2 inhibitor to reduce CM proliferation (however, ErbB2 also regulates EMT). We did not observe a significant difference in the number of extruding CMs in treated *snai1b* mutants compared with control (Author response image 2), while when we treated wild-type embryos with the ErbB2 inhibitor, we observed an increase in CM extrusion (data not shown). In addition, previous cell transplantation studies (Liu et al., 2010) analysed wild-type hearts with a few *erbb2* mutant CMs and did not report CM extrusion. Altogether, these data suggest that increased CM extrusion in *snai1b* mutants is not caused by increased CM proliferation or defective EMT. However, additional analysis using tools that specifically block CM proliferation versus EMT will be needed to further investigate these interesting questions.

**Author response image 2. sa2fig2:** The number of extruding CMs is not significantly different between DMSO and ErbB2 inhibitor treated *snai1b^-/-^
*embryos at 52 hpf. (*snai1b^-/-^* DMSO, n=23; *snai1b^-/-^
*ErbB2 inhibitor, n=35). Plot values represent means ± S.D.; p-values determined by Mann-Whitney *U* test. n, number of embryos.

The authors show that extruding cells do not seem to be dead or dying, and that a small number of CMs do extrude in wild type embryos. This raises the intriguing possibility that some amount of CM extrusion is necessary for normal development and that these cells may give rise to epicardial or other cell types. Live-imaging and lineage-tracing studies would inform whether the extrusion observed in mutant embryos is an enhancement of a normal morphogenetic process or an additional abnormal response to loss of Snai1 function.

We thank the reviewer for these interesting comments. We have now included a time-lapse spinning disk movie of the *snai1b* mutant hearts from 52 to 70 hpf, in which we observed CMs outside of the myocardial wall and they remained in the pericardial cavity for several hours (see our rebuttal to Major Point #2 in reviewer 2’s comment).

To test the hypothesis that CM extrusion is a normal process that gives rise to other cell types outside the myocardial wall, we performed a lineage-tracing experiment in wild-type embryos. We treated *Tg(myl7:cre^ERT2^);(-3.5ubb:loxP-EGFP-loxP-mCherry)* embryos with tamoxifen from 24 to 72 hpf, and imaged the larvae at 96 hpf. We found no switched (EGFP to mCherry) CM-derived cells in the pericardial cavity (Author response image 3). From our lineage tracing analysis, we believe that the extruded CMs do not contribute to other cardiac cells. Our time-lapse movies also show that in *snai1b* mutants, extruded CMs are not attached to and are positionally distant from the heart (Figure 1 —figure supplement 1I-K), thereby indicating that it is unlikely the extruded CMs give rise to epicardial or other cardiovascular cells.

**Author response image 3. sa2fig3:** Single-plane images of *Tg(myl7:cre^ERT2^);(-3.5ubb:loxP-EGFP-loxP-mCherry);(myl7:BFP-CAAX)* larvae at 96 hpf. No switched cells are found in the pericardial cavity. n=23. Scale bar: 20 µm. n, number of embryos.

One particularly interesting observation that was left unexplored was the identification of a second class of Snai1b mutants with defective heart looping. It isn't clear whether these embryos also display enhanced CM extrusion, or if there are other clearly aberrant cell behaviors. Furthermore, it would be very interesting to know whether there is any evidence that the defective looping is due to the same changes in cytoskeletal gene expression and protein organization observed in the class of Snai1b mutants that were detailed throughout the manuscript.

We thank the reviewer for these comments. We have now examined in more detail CM morphology in unlooped *snai1b* mutant hearts and have included some quantification in the revised manuscript. We found that the number of extruding CMs is similar in all *snai1b* mutants regardless of the looping phenotype. These new data are included in Figure 1 —figure supplement 1C-E. It will indeed be interesting to investigate whether changes in cytoskeletal gene and protein expression are also an underlying cause of the cardiac looping phenotype in *snai1b* mutants.

The authors suggest that Snai1b regulates Desmin in two ways: 1) overall expression levels, and 2) post-translationally to control its localization at cell junctions. Although the first claim is sufficiently supported, the second claim lacks experimental evidence. An alternative explanation is that overexpression of Desmin in response to loss of Snai1b leads to mislocalization independent of an interaction with Snai1b. This point could be clarified by examining Desmin localization in the desmb overexpression system. In addition, assaying for co-IP of Snai1b and Desmin could demonstrate a direct interaction between the two and better support a role for Snai1 in regulating post-translational localization of Desmin.

We thank the reviewer for these comments. We have now performed anti-Desmin immunostaining in *desmb* overexpressing CMs, and found that Desmin is enriched basally. This result suggests that an overabundance of Desmin can lead to its basal enrichment. However, whether Snai1b and Desmin interact at the protein level will need additional tools and analyses, and thus we have removed the corresponding sentence from the manuscript.

Although the authors convincingly show that Desmin accumulates with other contractile machinery at the basal side of extruding CMs in Snai1b muntants, additional evidence is needed to support a causal link between basal Desmin accumulation and extrusion. For instance, if knockdown or inhibition of Desmin prevents extrusion in the Snai1b mutants, the causal relationship would be much clearer.

We thank the reviewer for this suggestion. We used a *desmb* ATG morpholino to knock down *desmb* in *snai1b* mutants. Although we used a low concentration of the morpholino (0.5 ng), we surprisingly observed an increased number of extruding CMs in both wild types and *snai1b* mutants compared with standard control morpholino injections (Author response image 4). We hypothesize that the right balance of Desmin expression is needed to preserve myocardial wall integrity; too much or too little of Desmin increases CM extrusions. However, we cannot exclude the possibility that the effects observed are due to off-target effects of the morpholino. Due to these uncertainties, we did not include the *desmb* morpholino data in the manuscript.

**Author response image 4. sa2fig4:** The number of extruding CMs increases in *snai1b^+/+^
*and *snai1b^-/-^desmb* morphants compared with the respective control morphants at 52 hpf. (*snai1b^+/+^* control MO, n=5; *snai1b^+/+^ desmb* MO, n=8; *snai1b^-/-^* Control MO, n=5; *snai1b^-/-^ desmb* MO, n=9). Plot values represent means ± S.D.; p-values determined by Mann-Whitney *U* test. n, number of embryos..